# Strike a Pose: Relationships Between Infants’ Motor Development and Visuospatial Representations of Bodies

**DOI:** 10.3390/bs15081021

**Published:** 2025-07-28

**Authors:** Emma L. Axelsson, Tayla Britton, Gurmeher K. Gulhati, Chloe Kelly, Helen Copeland, Luca McNamara, Hester Covell, Alyssa A. Quinn

**Affiliations:** 1School of Psychological Sciences, College of Engineering, Science and Environment, University of Newcastle Australia, Callaghan, NSW 2308, Australia; 2School of Medicine and Psychology, College of Science and Medicine, Australian National University, Canberra, ACT 2601, Australia

**Keywords:** body representations, motor development, eye tracking, infant cognition, inversion effect

## Abstract

Infants discriminate faces early in the first year, but research on infants’ discrimination of bodies is plagued by mixed findings. Using a familiarisation novelty preference method, we investigated 7- and 9-month-old infants’ discrimination of body postures presented in upright and inverted orientations, and with and without heads, along with relationships with gross and fine motor development. In our initial studies, 7-month-old infants discriminated upright headless postures with forward-facing and about-facing images. Eye tracking revealed that infants looked at the bodies of the upright headless postures the longest and at the heads of upright whole figures for 60–70% of the time regardless of the presence of faces, suggesting that heads detract attention from bodies. In a more stringent test, with similarly complex limb positions between test items, infants could not discriminate postures. With longer trials, the 7-month-olds demonstrated a familiarity preference for the upright whole figures, and the 9-month-olds demonstrated a novelty preference, albeit with a less robust effect. Unlike previous studies, we found that better gross motor skills were related to the 7-month-olds’ better discrimination of upright headless postures compared to inverted postures. The 9-month-old infants’ lower gross and fine motor skills were associated with a stronger preference for inverted compared to upright whole figures. This is further evidence of a configural representation of bodies in infancy, but it is constrained by an upper bias (heads in upright figures, feet in inverted), the test item similarity, and the trial duration. The measure and type of motor development reveals differential relationships with infants’ representations of bodies.

## 1. Introduction

Faces and bodies are socially relevant visual objects that convey important information, such as a person’s identity, emotional state, gender, age, and intentions, and are therefore useful in aiding interactions with our social and physical environment ([45]; [64]; [70]). Faces and bodies are some of the most prominent classes of visual objects ([23]), and the neural perception of human faces and bodies differs from the perception of other categories because we ourselves possess a face and body ([29]; [50]; [64]; [71]). In adults, areas of the brain respond specifically to faces and bodies such as the occipital face area (OFA), the fusiform face area (FFA), the extrastriate body area (EBA), the fusiform body area (FBA), and the superior temporal sulcus (STS) ([45]; [48]). There is recent evidence of a similar activation in the FFA and EBA in infants ([38]). Infants’ neural and behavioural responses to faces in the first year of life have been studied extensively ([17]; [69]), but there has been far less research on infants’ responses to bodies. Bodies are also unique to other visual objects as we interact with the world via our own bodies, but the first year of life is characterised by rapid changes in motor development ([15]; [53]; [55]). Infants explore and interact with the world around them using their bodies to gather and integrate multisensory (i.e., visual, tactile, auditory and proprioceptive) information into coherent representations of the environment ([19]; [71]). Therefore, motor development might interact with infants’ visual representations of bodies, but few studies have investigated the influences of motor development on infants’ body representations.

## 2. Infants’ Discrimination of Bodies

Infants demonstrate sensitivity to faces (e.g., [60]), but it is unlikely that infants’ developing representations of people are restricted to heads. There are also many situations where infants’ view of heads is obscured, and they need to rely on bodies to obtain information about people. Initial findings on infants’ responses to bodies are suggestive of a more protracted form of development. There are also disparities in the age at which infants demonstrate sensitivity to bodies. From birth, newborn infants prefer typically configured faces to scrambled faces presented in static images (i.e., faces with features in atypical positions; [34]), but [63] ([63]) found that infants did not reliably discriminate static images of typical and scrambled bodies (i.e., bodies with limbs in atypical positions) until 18 months of age. With more life-like stimuli such as dolls, mannequins, real people, and images presented with human voices, infants discriminated bodies in the first year of life.

Other studies suggest infants can discriminate bodies in static images from a younger age. [28] ([28]) found larger event-related potentials (ERPs) in response to scrambled bodies than to typical bodies in 3-month-olds. [77] ([77]) found that 3.5-month-olds could discriminate typical and scrambled bodies, as well as bodies with proportionate and disproportionately sized body parts (i.e., stretched neck and torso, shortened legs). Interestingly, these findings were only seen with upright images rather than inverted images and suggest that infants as young as 3 months are sensitive to the structural configuration of human bodies in their typical orientation (but see [55]). [31] ([31]) found that 5- and 9-month-olds could discriminate body postures (i.e., with differing limb positions) with whole bodies but not with scrambled bodies or postures presented as isolated parts (i.e., without a torso, head, or limbs that do not change position). [31] ([31]) argued that this is evidence of infants’ holistic representation of bodies.

Newborns also demonstrate a preference for attractive over unattractive faces ([62]), but a preference for bodies based on attractiveness was not seen until 9 months of age by [30] ([30]), with a preference for unattractive bodies (based on adult ratings of body stimuli). Importantly, this was only found when heads were occluded and not with whole figures (bodies presented with heads). [30] ([30]) concluded that the preference for unattractive bodies was possibly due to a familiarity with a more typical body type.

A possible reason for the contrast in findings to [63]’s ([63]) initial studies is that their stimuli always contained faces which might have detracted infants’ attention from bodies. The more recent studies also presented fewer stimuli and test trials, with [77] ([77]) presenting only one pair of typical and scrambled bodies, [30] ([30]) presenting two pairs of attractive and unattractive bodies, and [31] ([31]) presenting one pair of postures in each condition type (e.g., one whole-body pair). [63] ([63]) presented six pairs of typical and scrambled bodies, and the greater number of trials and the variety in the body stimuli possibly introduced a higher cognitive demand.

## 3. Face and Body Inversion Effects

We typically see faces and bodies in a particular orientation. When discriminating pairs of inverted faces, adults are less accurate and slower than they are for pairs of upright faces; an effect that is stronger than that found for objects (e.g., houses and airplanes), suggesting that the effect is face-specific ([72]). A face inversion effect (FIE) has also been found with 4-month-old infants with looking time data ([69]), with 6- and 12-month-olds with ERP data ([5]; [18]), and 5- to 8-month-olds with haemodynamic brain responses ([44]).

Adults are also subject to body inversion effects (BIEs). [50] ([50]) found a BIE in adults’ discrimination of body postures that had a similar magnitude to the FIE, but they did not find an inversion effect with images of houses. The BIE has been replicated in subsequent studies with images of body postures (e.g., [4]; [73]) and body identities ([41]; [56]).

One explanation for inversion effects is the disruption of the global, holistic, or configural processing of the features, which is the simultaneous integration of the parts or features (such as the eyes, nose, and mouth) into a whole representation ([46]; [51]). One aspect of configural processing is first-order, which is the positional relationship between facial or bodily features (e.g., eyes above the nose and nose above the mouth for faces; and head above the arms and torso above the legs for bodies), an aspect that is important in the detection of faces and bodies ([20]). Second-order configural processing involves variations in the distances between features, which affects the recognition or discrimination between identities of faces ([20]) and between bodies in varying positions ([51]). Inversion disrupts observers’ first- and, in particular, second-order configural processing, and the effect typically occurs or is stronger with highly familiar categories ([16]; [20]). Infants are also sensitive to first- and second-order configural relations in faces in the first year of life ([13]; [58]; [67]). Infants are better at discriminating upright facial identities ([13]; [69]), typical and scrambled bodies ([77]), and bodies depicting emotions ([76]) compared to inverted versions, suggesting that upright faces and bodies are likely a familiar category for infants. However, [27] ([27]) found that differential brain responses to upright and inverted bodies appear later, at 14 months, than they do for upright and inverted faces. Furthermore, [55] ([55]) found that 5- to 14-month-old infants did not demonstrate a preference between bodies with proportionate or disproportionate body parts when presented as upright or inverted. Therefore, the evidence of the effects of orientation on infants’ representation of bodies is mixed and requires further investigation.

**Headless Bodies.** Surprisingly, the adult BIE disappears or is reduced when bodies are presented without heads (i.e., headless), but the BIE is found with bodies with missing limbs (i.e., arms or a leg), suggesting that heads are important in adults’ visual processing of the human bodies ([2]; [4], [3]; [56]; [73]). [73] ([73]) argued that face and body discrimination involves different cortical mechanisms and that the configural processing of faces is needed to induce the configural processing of bodies. Support for this was found by [4] ([4]), who found a BIE with forward-facing headless bodies but not about-facing headless bodies. [7] ([7]) also found that face-selective areas (FFA and OFA) had greater activation during the discrimination of upright whole figures, the most typical body type, while body-selective areas (EBA and FBA) had greater activity during the discrimination of both whole and headless figures, presented as upright and inverted. [2] ([2]) also found that when discriminating inverted body postures, adult participants focussed on the lower torso, but when discriminating upright postures, adults focussed on the heads and upper torso and even performed better at discriminating body postures when encouraged to focus on the heads.

## 4. Relationships with Motor Development

Given that we exist within a body, our representation of bodies is developed through not only visuospatial but also sensorimotor information ([11]; [61]), more specifically, through our visual and embodied experiences ([52]). Embodiment refers to the spatial awareness of one’s body, which is developed through proprioceptive feedback and through our body movements ([1]). From birth, infants are sensitive to multisensory information (e.g., visual, tactile, auditory, and proprioceptive) related to their own bodies, and this sensitivity increases when this input is synchronous ([24]). This is important because navigating the environment relies heavily on the integration of multisensory input to support social orientation and movement coordination ([14]). As the first year of life is also characterised by vast changes in infants’ motor development, infants’ motor skills could be linked to their ability to distinguish bodies. Familiarity is important to infants’ sensitivity to differences in stimuli (e.g., [27]), and as bodies reflect movement, their motor skills might relate to their increased visual experience of bodies ([35]). However, the evidence for this is unclear.

[15] ([15]) investigated the relationship between 6- and 15-month-old infants’ gross and fine motor development and infants’ discrimination of static images of typical and scrambled bodies but found no significant relationships. More recently, [55] ([55]) found that 5- to 14-month-old infants’ gross motor development (e.g., ability to crawl or walk) was also not related to infants’ discrimination of static images of typical bodies and bodies with disproportioned body parts. However, [53] ([53]) found that 8-month-old infants’ greater fine motor development was significantly related to significantly longer looking at biologically impossible arm and hand movements, suggesting that greater familiarity with arm and hand movements contributed to infants’ recognition of impossible movements. Other studies where infants’ capacity to move is experimentally manipulated, such as using ‘sticky mittens’ to allow infants to pick up objects before they are capable (e.g., [65]), and engaging in reflexive walking motion ([54]), demonstrated infants’ better capacity to visually discriminate biological body motion.

Therefore, it is surprising that there is minimal evidence of a relationship between infants’ visual responses to bodies and their motor development. This could be due to the use of static images or to the use of non-veridical stimuli, namely scrambled bodies ([15]) and bodies with stretched limbs ([55]). Infants’ motor development may relate to more familiar stimuli ([53]), like body postures. Body postures provide information about a person’s actions, intentions, and emotional state ([50]; [78]). Infants experience their own embodied bodily positions ([68]) and attend to and imitate others’ body movements ([35]). We investigated relationships between infants’ gross and fine motor skills and their ability to discriminate body postures. Infants’ developing use of their bodies might contribute to their developing capacity to visually discriminate and interpret the bodily movements of others in their environment, ultimately contributing to their social–cognitive development.

## 5. Current Study

We tested 7- and 9-month-old infants’ ability to discriminate body postures in a series of studies (see Figure 1). We used a familiarisation novelty preference method, which involved familiarising infants with one posture and then presenting the same posture with a novel posture. As infants tend to prefer novelty, they typically look longer at a novel image if they can discriminate between two test images ([21]; but see [25]; [32]). The postures were presented in upright and inverted orientations and with heads (whole figures) and without heads (headless). [77] ([77]) found that infants only discriminated typical and scrambled bodies when presented upright. [31] ([31]) found that infants only discriminated postures presented as whole figures rather than isolated parts. [30] ([30]) found that 9-month-olds preferred attractive and unattractive bodies only when heads were occluded, but with a discrimination test, 6-month-olds distinguished the bodies. In contrast, adults’ typical visual processing of bodies is disrupted with headless stimuli (e.g., [7]). Given the above findings, we predicted that 7-month-old infants would discriminate the upright whole and headless postures. Unlike previous studies on infants’ discrimination of bodies, we used eye tracking to measure infants’ direction and duration of gaze to the heads, bodies, and feet of the images. In the first study (1A), 7-month-old infants discriminated the headless upright postures only. When viewing the whole figures, they demonstrated a strong interest in the heads, which potentially detracted attention from the bodies (see Appendix A). This called for a follow-up study (1B) with images of people from behind to provide an ecologically valid way to remove facial information. Infants again showed a heavy interest in the heads and discriminated postures in the upright headless condition only. One issue with these initial studies was that the familiar posture was typically neutrally positioned (i.e., with straight arms and legs), while the limbs of the novel postures were in a variety of positions. Infants might have demonstrated novelty preferences because the novel postures were more attention-grabbing and not just because they were novel. Therefore, another study was performed with the limbs of both figures of each pair of postures in a variety of positions (2A). With this study, 7-month-olds did not discriminate the postures. Another study with longer trial durations and the addition of 9-month-olds was performed (2B). We also included a measure of gross and fine motor development to assess the role of infants’ developing use of their bodies in their ability to discriminate body postures. We predicted that stronger gross and fine motor development would be related to infants’ stronger capacity to discriminate the body postures.

## 6. Study 1A—7-Month-Old Infants’ Discrimination of Body Postures

### 6.1. Method

#### 6.1.1. Participants

Infants were recruited via advertisements placed on social media and flyers distributed to childcare centres and cafes. This study was approved by the human research ethics committee at The Australian National University (Protocol number 2015/183). Data and Appendix A are available on the Open Science Framework (https://osf.io/epc5z/ URL accessed on 3 January 2025). The final sample consisted of 24 healthy, full-term 7-month-old infants (age range: 7 months 0 days—7 months 30 days, *M* age = 7 months 12 days, *SD* = 10 days; 13 female, 11 male). One infants’ data was omitted due to drowsiness and fussiness. Using G*Power, version 3.1.9.2 ([22]), a power analysis was conducted based on the effect size found by [31] ([31]) (*d* = 1.38, α = 0.05, two-tailed; 1 − β = 0.80). This indicated that only 7 participants were required. However, as [31]’s ([31]) effect size was extremely strong, we aimed for a standard 20–24 infants, a size typically used in similar studies (e.g., [30]; [49]; [75], [77]). This was the target sample size in the remaining studies. The mothers’ mean age was 31.40 years (*SD* = 4.17), and 47% had completed a post-graduate degree, 46% a bachelor’s degree or diploma, 5% high school, and 2% junior high school. The fathers’ mean age was 38.05 years (*SD* = 17.36), and 23% had completed a post-graduate degree, 62% a bachelor’s degree or diploma, 9% high school, and 6% junior high school. The caregivers identified their ethnic background as Australian (63%), Asian (6%), European (6%), New Zealand (4%), American (North and South) (4%), and mixed (13%).

#### 6.1.2. Materials

**Eye Tracking.** An EyeLink 1000 eye tracker (SR-Research.com), with a 16 mm lens, was used to record the corneal reflections and pupils at a sampling rate of 500 Hz and with 0.5° spatial accuracy. The ‘Remote’ setting was used, which meant infants could move within a spatial area of 22 × 18 × 20 cm with the help of a high-contrast sticker (5 mm diameter) on the infant’s forehead. The eye-tracking camera was positioned directly in front of and beneath a 24-inch Dell computer monitor with a resolution of 1920 × 1080 pixels and 60 Hz refresh rate. The experiment was created using Experiment Builder software (version 2.3.1), and the eye-tracking data was extracted using Data Viewer software (version 3.1.1, SR-Research.com).

**Stimuli.** The stimuli for this study were images of 8 adult males sourced from Shutterstock.com. Male figures were used all with similar characteristics, including short hair, little to no facial hair, neutral facial expressions, form-fitting clothing (i.e., trousers or jeans and a shirt), and no attention-grabbing features (e.g., jewellery). Each male appeared as a pair with two different postures (see Figure 2). Within each pair, one posture was presented as the familiar posture and the other as the novel. Postural adjustments were made by adjusting the arms and/or legs using Adobe Photoshop CS6. The familiar postures were typically in a neutral position and the novel postures had more variation in limb positions with either a leg, arm, or leg and arm in differing positions to the familiar posture (see Figure 2). Four versions of each pair were created to correspond with each condition, whole figure upright (body with head), whole figure inverted, headless upright (body without head), and headless inverted, leading to a total of 32 images. For the headless conditions, the heads and necks of the images were erased from the top of the shirt line using Adobe Photoshop CS6. For the inverted condition, the images were rotated 180°. The figures were positioned in a forward-facing stance, and, importantly, the heads within each pair of whole-figured bodies were identical—only the postures differed. The average size of the whole-figured bodies was 152.28 × 272.47 mm (299.87 × 675.03 pixels; 14.45° × 25.59° visual angle at 60 cm from the screen). The average size of headless bodies was 152.28 × 234.33 mm (299.88 × 580.52 pixels; 14.45° × 22.10°). For the eye-tracking analyses (see Appendix A), polygonal areas of interest (AOIs) were placed around the heads, bodies, and feet in each figure (see Figure 2). Looking time proportions to each part (heads, bodies, feet) were calculated by dividing the total looking time at the area of interest (e.g., body) by the total time spent looking at the entire figure.

#### 6.1.3. Procedure

Prior to commencing eye tracking, the experimenters played with the infant to ensure they were comfortable in the new laboratory environment, alert, and ready to view the images. Prior to commencing, the caregivers were instructed to hold their infants in an upright position and to allow their child to independently view the images and avoid directing their attention. Infants sat on their caregiver’s lap approximately 55 cm from the eye-tracking camera and 60 cm from the display screen. Infants viewed an animated movie featuring abstract shapes while the experimenter ensured that the camera and display screen were at an ideal distance and height to the infant. Calibration was conducted by presenting animated attention-getters (spinning geometric shapes accompanied by audio) at five positions across the display screen (centre, top, bottom, left, and right sides). Prior to the presentation of each image, an attention-getting animation appeared in the centre of the screen to ensure children were centrally fixated. Once infants fixated on the attention-getter for 300 ms, infants saw one posture from a given pair. To familiarise infants to the first posture, it remained on the screen until 5000 ms of looking had accumulated. This was to ensure the participant was familiar with the posture. The 5000 ms duration was based on pilot testing where infants’ attention waned after this time. Two test trials followed consisting of the familiarised posture and the same person in a novel posture. Each pair was presented for 5000 ms, and in the subsequent test trial the side position of the pairs was switched to attempt to address any side biases. Infants saw each condition once, and the presentation of the conditions was counterbalanced across infants as was the side at which the novel posture first appeared. The position of the novel posture alternated sides on subsequent trials. Infants saw a different pair (male identity) in each condition, and the presentation of a given pair (of 8) was counterbalanced across the conditions across infants. As there were four conditions, each pair was seen by half of the participants, and the pairs were presented an equal number of times across the infants. Participants received an AUD 10 gift card and storybook as a token of appreciation.

### 6.2. Design and Analysis

This study has a repeated measures experimental design with four conditions: whole figure upright, whole figure inverted, headless upright, and headless inverted (see Figure 2). For each condition, novelty preference scores for the novel postures in each test trial were calculated by dividing the total looking time to the new posture by the total looking time to the familiar and new posture. Infants’ ability to discriminate bodies was assessed by comparing infants’ novelty preferences to chance (i.e., 0.50 due to the presence of two postures) using two-tailed one-sample *t*-tests.

### 6.3. Results

#### Novelty Preferences Compared to Chance

First, any extreme novelty preference scores (>0.95 or <0.05) were removed (five in the whole figure upright condition; four in the whole figure inverted condition; seven in the headless upright condition; and seven in the headless inverted condition). This is in accordance with the standard practice of the familiarisation novelty preference method to ensure that novelty preferences are based on infants having attended to each test image for a necessary minimum amount of time ([49]). Additionally, to ensure the analyses were not affected by extreme values, novelty preference scores associated with *z*-scores beyond −2/+2 were replaced with the nearest score within −2/+2 (three in the whole figure inverted condition, one in the headless upright condition). Some trials also had no looking (two in the whole figure upright condition; two in the headless inverted condition).

**Novelty Preferences Test Trial One.** The novelty preferences were not significantly different to chance in any of the conditions (whole figure upright: *t*(20) = 1.64, *p* = 0.117, *d* = 0.36; whole figure inverted: *t*(22) = −0.01, *p* = 0.990, *d* = −0.00; headless upright: *t*(19) = 1.51, *p* = 0.147, *d* = 0.34; and headless inverted: *t*(18) = −0.41, *p* = 0.690, *d* = −0.09; see Figure 3).

**Novelty Preferences Test Trial Two.** The novelty preferences were only significantly higher than chance in the headless upright condition with a medium effect size, *t*(20) = 3.57, *p* = 0.002, and *d* = 0.78. This effect remained significant following adjustments for multiple tests (Benjamini–Hochberg FDR adjustments for multiple comparisons across four conditions, α < 0.0125). The remaining novelty preference scores were not significantly different to chance (whole figure upright: *t*(17) = 1.18, *p* = 0.254, *d* = 0.28; whole figure inverted: *t*(20) = 1.29, *p* = 0.213, *d* = 0.28; headless inverted: *t*(18) = −0.66, *p* = 0.516, *d* = 0.15; see Figure 3).

### 6.4. Discussion

Here, 7-month-olds discriminated body postures in the upright headless condition only and only in the second test trial. The eye-tracking data revealed that a large proportion of time (~70%) was spent looking at the heads of the whole figures (relative to bodies and feet, see Appendix A). With headless bodies, infants spent a longer time looking at the bodies (~90%). Therefore, infants seemed to display an attentional bias towards heads. When distracted by heads, infants spend less time looking at bodies, which may consequently affect their capacity to discriminate bodies. As facial features might attract infants’ attention, we ran this study again with images of people viewed from behind as a way to remove facial features and assess if infants would discriminate postures in the whole-figure condition.

## 7. Study 1B—7-Month-Old Infants’ Discrimination of About-Facing Body Postures

All elements of this study were the same as Study 1A, aside from the presentation of about-facing figures with a new sample of 7-month-olds.

### 7.1. Method

#### 7.1.1. Participants

The final sample consisted of 24 full-term 7-month-old infants (*M* = 7 months 12 days, *SD* = 9 days; range = 7 months 0 days to 7 months 29 days; 10 female). A further 5 infants were tested, but their data were excluded due to difficulties tracking the participants’ eye (*n* = 2) and excessive movement (*n* = 3). The mothers’ mean age was 31.59 years (*SD* = 5.04), and 31% had completed a post-graduate degree, 62% a bachelor’s degree or diploma, 7and % high school. The fathers’ mean age was 32.95 years (*SD* = 4.97), and 41% had completed a post-graduate degree, 41% a bachelor’s degree or diploma, 7% high school, and 11% junior high school. The caregivers identified their ethnic background as Australian (68%), Asian (9%), European (13%), New Zealand (4%), and mixed (9%).

#### 7.1.2. Materials and Procedure

**Stimuli.** All design elements of the stimuli were the same as Study 1A, but here all images were of people from behind (see Figure 4). Eight pairs of photographs of about-facing adult male figures were sourced from Shutterstock.com. Due to limited availability of the original images viewed from behind, 6 of the identities were new. The average size of the whole-figured bodies was 158.66 × 272.04 mm (312.44 × 666.95 pixels; 15.05° × 25.55° visual angle at 60 cm from the screen). The average size of headless bodies was 158.66 × 238.36 mm (312.44 × 584.37 pixels; 15.05° × 22.47°). The procedure was the same as Study 1A.

### 7.2. Results

#### Novelty Preferences Compared to Chance

Extreme novelty preference scores (>0.95 or <0.05) were removed (five in the whole figure upright condition, six in the whole figure inverted condition; and five in the headless inverted condition), and any extreme novelty preferences (*z*-scores beyond −2/+2) were replaced with the nearest score within −2/+2 (one in the whole figure upright condition; one in the whole figure inverted condition; one in the headless upright condition; one in the headless inverted condition). Some trials had no looking (one in the whole figure inverted condition; two in the headless upright condition).

**Novelty Preferences Test Trial One.** The novelty preferences were significantly different to chance in the headless upright condition only and with a medium effect size, *t*(22) = 2.98, *p* = 0.007, and *d* = 0.62. This effect remained significant following adjustments for multiple tests (Benjamini–Hochberg FDR adjustments for multiple comparisons across four conditions, α < 0.0125). The novelty preferences were not significantly different to chance in the remaining conditions (whole figure upright: *t*(20) = 1.11, *p* = 0.278, *d* = 0.24; whole figure inverted: *t*(20) = 0.54, *p* = 0.596, *d* = 0.12; headless inverted: *t*(21) = 0.30, *p* = 0.769, *d* = 0.06; see Figure 5).

**Novelty Preferences Test Trial Two.** The novelty preferences did not differ significantly to chance in all of the conditions (whole figure upright: *t*(21) = 1.16, *p* = 0.261, *d* = 0.25; whole figure inverted: *t*(19) = 1.05, *p* = 0.308, *d* = 0.23; headless upright condition *t*(22) = 0.28, *p* = 0.782, *d* = 0.06; headless inverted: *t*(20) = 0.47, *p* = 0.642, *d* = 0.1; see Figure 5).

### 7.3. Discussion

Like Study 1A, infants demonstrated an ability to discriminate postures in the headless upright condition, but unlike Study 1A, this was in the first test trial only. Proportional looking to the upright heads was 59% with the about-facing figures, which is less than it was with the forward-facing figures (72%), suggesting that the attention to the heads was reduced with about-facing figures (see Appendix A for more). However, infants still spent more than half the time viewing the about-facing heads even though the faces were not visible. This might, however, explain why the novelty preference was significant in the first test trial. Infants likely spent more time viewing the body, allowing infants to recognise the novel posture in the first test trial.

A limitation of Studies 1A and 1B is that the first posture was neutral with straight arms and legs, and the novel posture had varied arm and leg positions potentially making it more attention-grabbing. To address this confound, Study 1A was conducted again with each figure in a pair having similarly varied postures.

## 8. Study 2A—7-Month-Old Infants’ Discrimination of Similarly Varied Body Postures; 5 s Trial Durations

All elements of this study were the same as Study 1A, aside from the figures in each test pair having similarly varied limb positions, but the postures remained different. This also increased the similarity of the width of the test pairs. A new sample of 7-month-olds were tested.

### 8.1. Method

#### 8.1.1. Participants

A total of 25 seven-month-old infants were included in the final analyses (*M* = 7 months 16 days, *SD* = 7 days; range: 7 months 3 days to 7 months 29 days). A further 8 participants were tested but their data was excluded due to excessive movement and/or inattentiveness (*n* = 5) and smiling and squinting obscuring the pupil (*n* = 3). The mothers’ mean age was 32.55 years (*SD* = 4.04), and 47% had completed a post-graduate degree, 50% a bachelor’s degree or diploma, and 3% high school. The fathers’ mean age was 36.38 years (*SD* = 12.17), and 22% had completed a post-graduate degree, 53% a bachelor’s degree or diploma, and 25% high school. The caregivers identified their ethnic background as Australian (69%), Asian (11%), European (3%), New Zealand Māori (2%), American (5%), and mixed/other (11%).

#### 8.1.2. Stimuli and Procedure

The same male identities as Study 1A were used except this time the postures had a similar degree of variety in the limb positions between the familiar and novel image, and this resulted in a similar width. That is, instead of the familiar stimuli having largely straight arms and legs, they had a variety of positions like the novel stimuli, while also ensuring they differed in positions to the respective image within a pair (see Figure 6). The procedure remained the same as the previous studies.

### 8.2. Results

#### Novelty Preferences Compared to Chance

Extreme novelty preference scores (>0.95 or <0.05) were removed (two in the whole figure upright condition; five in the whole figure inverted condition; seven in the headless upright condition; eight in the headless inverted condition), and any extreme novelty preferences (*z*-scores beyond −2/+2) were replaced with the nearest score within −2/+2 (one in the whole figure upright condition; two in the whole figure inverted condition; two in the headless inverted condition). Some trials also had no looking (one in the whole figure upright condition; two in the headless upright condition).

**Novelty Preferences Test Trial One.** The novelty preferences did not differ significantly to chance in any of the conditions (whole figure upright: *t*(22) = −0.08, *p* = 0.935, *d* = −0.02; whole figure inverted: *t*(21) = 1.56, *p* = 0.134, *d* = 0.33; headless upright: *t*(21) = 1.23, *p* = 0.232, *d* = 0.26; headless inverted: *t*(19) = 1.58, *p* = 0.130, *d* = 0.35; see Figure 7).

**Novelty Preferences Test Trial Two.** The same was found in Test Trial Two, with none of the novelty preferences differing significantly to chance (whole figure upright: *t*(23) = 0.12, *p* = 0.905, *d* = 0.02; whole figure inverted: *t*(22) = 0.00, *p* = 0.996, *d* = 0.00; headless upright condition *t*(18) = −0.19, *p* = 0.848, *d* = −0.04; headless inverted: *t*(19) = 0.95, *p* = 0.355, *d* = 0.21; see Figure 7).

### 8.3. Discussion

Presenting similarly complex stimuli likely introduced a level of difficulty that led to the 7-month-olds not demonstrating an ability to discriminate the body postures. This is either because of a lack of sensitivity to differences in body postures or due to constraints of the experiment. It is possible that infants require longer trial durations with these stimuli. It is also possible that the increased similarity in the body postures is too difficult for 7-month-old infants to discriminate. Therefore, a follow-up study was conducted with longer trials, and we added an older age group, 9-month-olds, to test if developmental constraints contribute to the difficulty in discriminating the body postures.

## 9. Study 2B—7- and 9-Month-Old Infants’ Discrimination of Similarly Varied Body Postures with 8 s Trials: Associations with Motor Development

This study was approved by the human research ethics committee at the University of Newcastle (protocol number H-2022-0069). All elements of this study were the same as Study 2A; however, the trial duration was increased to 8 s, and both 7- and 9-month-olds were tested. Increasing the similarity of the test images likely increased the difficulty, so an older age group was warranted. We also included a measure of gross and fine motor development to attempt to determine if infants’ developing use of their own bodies was associated with a capacity to discriminate body postures.

### 9.1. Method

#### 9.1.1. Participants

A total of 53 participants were included in the final analyses, with 27 infants in the 7-month age group (*M* = 7 months 12 days, *SD* = 8.80 days; range: 7 months 0 days to 7 months 29 days) and 26 in the 9-month age group (*M* = 9 months 14 days, *SD* = 9.69 days; range: 9 months 0 days to 9 months 30 days). A further 11 participants were tested, but their data was excluded due to fussiness and inattentiveness (7-month-olds: *n* = 6; 9-month-olds: *n* = 2) and smiling and squinting obscuring the camera’s ability to capture the pupil and gaze (7-month-olds: *n* = 2; 9-month-olds: *n* = 1). The mothers’ mean age was 35.49 years (*SD* = 10.06), and 47% had completed a post-graduate degree, 46% a bachelor’s degree or diploma, 5% high school, and 2% junior high school. The fathers’ mean age was 37.34 years (*SD* = 11.88), and 23% had completed a post-graduate degree, 62% a bachelor’s degree or diploma, 9% high school, and 6% junior high school. The caregivers identified their ethnic background as Australian (85%), Indigenous Australian (3%), Asian (5%), European (3%), New Zealand Māori (2%), and American (2%).

#### 9.1.2. Materials

**Eye Tracking.** There was a change in institution for the first author to University of Newcastle, Australia, leading to the use of an upgraded version of the eye tracker to an EyeLink 1000 Plus (SR-Research.com). The display screen was attached to a moveable ‘arm-mount’ with the eye-tracking camera (16 mm lens) positioned beneath the display screen (25-inch Dell monitor with a resolution of 1920 × 1080 pixels and 60 Hz refresh rate). As with Study 1A, infants sat on a caregiver’s lap approximately 55 cm away from the camera and 60 cm from the screen (see https://www.sr-research.com/eyelink-1000-plus/ for example image of the set-up, accessed on 5 June 2025). The remote setting was used, which allows for movement in an area of 40 × 40 × 15 cm along with the help of a high contrast sticker (20 mm diameter) on the child’s forehead to aid in the detection of the child’s eye. Looking was sampled at 1000 Hz using monocular tracking with 0.5° spatial accuracy. The experiment was created using Experiment Builder software (version 2.3.1), and data was extracted using Data Viewer software (4.2.1).

**Stimuli.** These were the same as Study 2A.

The Ages and Stages Questionnaire-3 (ASQ-3; [66]). This is a brief standardised questionnaire measuring five developmental domains: gross motor, fine motor, communication, problem solving, and personal–social development. Each developmental domain has six questions answered with a three-point scale: yes = 10, sometimes = 5, and not yet = 0. Scores indicate whether development is on schedule (35–50), whether further activities and monitoring are needed (score 25–45), or whether further assessment is required (scores below 20–30). Different sets of questions are appropriate for different ages, with the 8-month questionnaire suitable for those aged 7 months 0 days to 8 months 30 days; and the 9-month questionnaire suitable for infants aged 9 months 0 days to 9 months 30 days. Example gross motor items include, ‘If you hold both hands just to balance your baby, does he support his own weight while standing?’ and ‘When sitting on the floor, does your baby sit up straight for several minutes without using her hands for support?’ Example fine motor items include, ‘Does your baby pick up a small toy with only one hand?’ and ‘Does your baby pick up a small toy with the tips of his thumb and fingers?’ Internal consistency for the ASQ-3 is moderate to excellent (Cronbach’s α = 0.51–0.87), test–retest reliability is very good (ICCs = 0.75–0.82), inter-rater reliability is moderate (ICCs = 0.43–0.69), and concurrent validity is high ([40]; [59]; [66]).

#### 9.1.3. Procedure

The same procedure as the previous studies was followed except that the familiarisation trial was now at 8 s of cumulative looking, and the test trials were 8 s each. This was based on pilot testing with 10 s, where infants had difficulties reaching 10 s of cumulative looking.

**Analyses.** Linear mixed-effects models were performed to compare the novelty preferences across conditions (whole figure upright, whole figure inverted, headless figure upright, headless figure inverted), across age groups, and across trials, and gross and fine motor scores were included to assess their relationships with infants’ novelty preference scores. Follow-up moderation analyses were conducted with the participants’ gross and fine motor raw scores assessed at low (1 *SD* below the sample mean), medium (within 1 *SD* of the sample mean), and high levels (1 *SD* above the sample mean) in relation to the comparisons between the novelty preference score for the upright and inverted conditions. All analyses were conducted using jamovi 2.3.28.0 software with the added GAMLj module 3.3.8.

### 9.2. Results

#### 9.2.1. Novelty Preferences Compared to Chance

First, any extreme novelty preference scores (>0.95 or <0.05) were removed (7-month-olds: five in the whole figure upright condition; three in the whole figure inverted condition; two in the headless upright condition; eight in the headless inverted condition; 9-month-olds: one in the whole figure upright condition; five in the whole figure inverted condition; two in the headless upright condition; one in the headless inverted condition). Scores associated with *z*-scores beyond −2/+2 were replaced with the nearest score within −2/+2 (7-month-olds: three in the whole figure upright condition; one in the headless upright condition; 9-month-olds: two in the whole figure upright condition; two in the headless upright condition; one in the headless inverted condition). Some trials also had no looking (7-month-olds: four in the whole figure upright condition; two in the whole figure inverted condition; 9-month-olds: two in the headless inverted condition).

**Novelty Preferences Test Trial One.** The 7-month-olds’ novelty preferences were not significantly different to chance in any of the conditions (whole figure upright: *t*(22) = 0.85, *p* = 0.404, *d* = 0.18; whole figure inverted: *t*(24) = 0.93, *p* = 0.364, *d* = 0.19; headless upright: *t*(25) = 0.01, *p* = 0.991, *d* = 0.01; headless inverted: *t*(22) = 0.79, *p* = 0.437, *d* = 0.17; see Figure 8). In Test Trial One, the 9-month-olds’ novelty preferences were significantly greater than chance in the whole figure upright condition with a small effect size, *t*(25) = 2.12, *p* = 0.044, *d* = 0.42; see Figure 8). However, this was non-significant following adjustments for multiple tests (Benjamini–Hochberg FDR adjustments for multiple comparisons across four conditions, α > 0.0125). The 9-month-olds’ novelty preferences were not significantly different to chance in the remaining conditions (whole figure inverted: *t*(23) = −0.50, *p* = 0.619, *d* = −0.10; headless upright: *t*(25) = 0.74, *p* = 0.464, *d* = 0.15; headless inverted: *t*(24) = −0.97, *p* = 0.341, *d* = −0.19; see Figure 8).

**Novelty Preferences: Test Trial Two**. The 7-month-olds’ novelty preference was significantly less than chance in the whole figure upright condition with a small effect size: *t*(20) = −2.19, *p* = 0.040, and *d* = −0.48. However, this was non-significant following adjustments for multiple tests (Benjamini–Hochberg FDR adjustments for multiple comparisons across four conditions, α > 0.0125). The novelty preferences were not significantly different to chance in the remaining conditions (headless inverted: *t*(22) = −1.39, *p* = 0.177, *d* = −0.29; headless upright: *t*(25) = 1.36, *p* = 0.185, *d* = 0.27; whole figure inverted: *t*(23) = −1.05, *p* = 0.304, *d* = −0.22). In Test Trial Two, the 9-month-olds’ novelty preferences were not significantly different to chance in all of the conditions (headless inverted: *t*(23) = 0.09, *p* = 0.933, *d* = 0.02; headless upright: *t*(23) = −0.26, *p* = 0.797, *d* = −0.05; whole figure inverted: *t*(23) = 1.58, *p* = 0.130, *d* = 0.33; whole figure upright: *t*(24) = −1.86, *p* = 0.075, *d* = −0.37; see Figure 9).

#### 9.2.2. Comparison of Novelty Preferences Across Age, Condition, Test Trial, and Relationships with Gross and Fine Motor Development

A linear mixed model (LMM) analysis was performed comparing the novelty preferences across the age groups (7 months, 9 months), conditions (repeated factors: whole figure upright vs. whole figure inverted, headless upright vs. headless inverted), test trials (first, second), and relationships with gross motor and fine motor development, along with the following interactions: the age by condition, age by gross motor, age by fine motor, age by test trial, condition by test trial, age by condition by gross motor, age by condition by fine motor, and age by condition by test trial (see Appendix A for fixed effect parameter estimates in Appendix A). Gross and fine motor scores were significantly negatively skewed (gross motor: Shapiro–Wilk’s *W* = 0.95, *p* = 0.036; fine motor: Shapiro–Wilk’s *W* = 0.86, *p* < 0.001). Attempts to transform them (reversed and log-transformed) did not improve the distribution. However, the normality tests within the LMM analyses revealed that the residuals were normally distributed, Shapiro–Wilk’s *W* = 0.99 and *p* = 0.058 (and all *p*s were > 0.05 for the follow-up moderation analyses within each test trial). Therefore, the original scores were used. The random effect of participants was non-significant, *LRT*(1) = 0.01 and *p* = 0.999 (random intercept variance = 0.001 *SD* = 0.001), and the intra-class correlation (*ICC*) was 0.01, suggesting minimal variability across participants. All effects and interactions were non-significant aside from the age by condition by test trial interaction, *F*(3,357) = 5.00 and *p* = 0.002 (see Appendix A for remaining non-significant main effects and interactions). This is explained by a difference in the novelty preferences between the two age groups for headless upright and headless inverted conditions across the two test trials. This is also explained by a difference between the two age groups for the novelty preferences in the whole figure upright and whole figure inverted conditions across the two test trials (see Appendix A for the fixed effects parameter estimates and Figure 8 and Figure 9). To clarify this, a follow-up simple effects comparison revealed that for the 7-month-olds only, novelty preferences in the second test trial were significantly greater in the headless upright condition (*M* = 0.55, *SE* = 0.04) compared to the headless inverted condition (*M* = 0.45, *SE* = 0.04), *t*(357) = −2.00 and *p* = 0.047, but not in the first test trial, *t*(357) = 0.47 and *p* = 0.462 (headless upright: *M* = 0.50, *SE* = 0.04; headless inverted: *M* = 0.54, *SE* = 0.04; see Figure 8 and Figure 9). However, this was non-significant following adjustments for multiple tests (Benjamini–Hochberg FDR adjustments for multiple comparisons across two conditions, α > 0.025). For the 9-month-olds only, novelty preferences in the second test trial were significantly greater in the whole figure inverted condition (*M* = 0.61, *SE* = 0.04) compared to the whole figure upright condition (*M* = 0.45, *SE* = 0.04), *t*(357) = 2.72 and *p* = 0.007, but not in the first test trial, *t*(357) = −1.85 and *p* = 0.065 (whole figure upright: *M* = 0.55, *SE* = 0.04; whole figure inverted: *M* = 0.45, *SE* = 0.04; see Figure 8 and Figure 9; see Appendix A for remaining non-significant effects). This remained significant following adjustments for multiple tests (Benjamini–Hochberg FDR adjustments for multiple comparisons across two conditions, α < 0.025).

#### 9.2.3. Gross Motor Moderation Analyses

Follow-up moderation analyses were performed with the relationship between novelty preferences and gross motor scores as the moderated variable at low, medium, and high levels within each comparison of orientation for the headless and whole-figure conditions (upright and inverted headless and whole figures) in each test trial.

**Test Trial One.** The relationships between gross motor and novelty preferences for each orientation comparison were non-significant for both the 7- and 9-month-olds (see Figure 10 and Appendix A).

**Test Trial Two.** The 7-month-olds with high levels of gross motor development (≥53) had significantly higher novelty preferences in the headless upright condition (*M* = 0.57, *SE* = 0.05) than in the headless inverted condition (*M* = 0.37, *SE* = 0.06; see Figure 11 and Appendix A). This effect remained significant following adjustments for multiple tests (Benjamini–Hochberg FDR adjustments for multiple comparisons across three levels, α < 0.0167). The 9-month-olds with low (≤18) and average levels (19 to 52) of gross motor development had significantly higher novelty preferences in the whole figure inverted condition (low gross motor development: *M* = 0.57, *SE* = 0.05; average gross motor development: *M* = 0.61, *SE* = 0.04) compared to the whole figure upright condition (low gross motor development: *M* = 0.41, *SE* = 0.05; average gross motor development: *M* = 0.46, *SE* = 0.04; see Figure 11 and Appendix A). This effect remained significant following adjustments for multiple tests (Benjamini–Hochberg FDR adjustments for multiple comparisons across three levels, α < 0.0167). These findings suggest that gross motor development contributes to infants’ discrimination of bodies.

#### 9.2.4. Fine Motor Moderation Analyses

The relationship between novelty preferences and fine motor development at low, medium, and high levels within each orientation comparison for the headless and whole-figure conditions (upright and inverted headless and whole figures) was also assessed.

**Test Trial One.** None of the relationships were significant for the 7-month-olds (see Figure 12 and Appendix A). The 9-month-olds with low levels of fine motor development (≤39) had significantly higher novelty preferences in the whole figure upright condition (*M* = 0.60, *SE* = 0.05) than in the whole figure inverted condition (*M* = 0.42, *SE* = 0.06; see Figure 12 and Appendix A). However, this was non-significant following adjustments for multiple tests (Benjamini–Hochberg FDR adjustments for multiple comparisons across three levels, α > 0.0167).

**Test Trial Two.** The 7-month-olds with low levels of fine motor development (≤39) had significantly higher novelty preferences in the headless upright condition (*M* = 0.60, *SE* = 0.05) than in the headless inverted condition (*M* = 0.44, *SE* = 0.05; see Figure 13 and Appendix A). However, this was non-significant following adjustments for multiple tests (Benjamini–Hochberg FDR adjustments for multiple comparisons across three levels, α > 0.0167). Also in Test Trial 2, the 9-month-olds with low levels of fine motor development (≤39) had significantly higher novelty preferences in the headless inverted condition (*M* = 0.59, *SE* = 0.06) than in the headless upright condition (*M* = 0.39, *SE* = 0.06; see Figure 13). However, this was non-significant following adjustments for multiple tests (Benjamini–Hochberg FDR adjustments for multiple comparisons across three levels, α > 0.0167). The 9-month-olds with low (≤39) and average levels (40 to 59) of fine motor development also had significantly higher novelty preferences in the whole figure inverted condition (low fine motor development: *M* = 0.67, *SE* = 0.06; average fine motor development: *M* = 0.61, *SE* = 0.04) than in the whole figure upright condition (low fine motor development: *M* = 0.43, *SE* = 0.06; average fine motor development: *M* = 0.46, *SE* = 0.04). This effect remained significant following adjustments for multiple tests (Benjamini–Hochberg FDR adjustments for multiple comparisons across three levels, α < 0.0167). These findings suggest that fine motor development contributes to infants’ discrimination of bodies but at lower levels of fine motor development.

### 9.3. Discussion

Infants demonstrated an ability to discriminate postures with the upright whole figures, with 9-month-old infants demonstrating a significant preference for the novel posture in the first test trial and the 7-month-olds demonstrating a significant preference for the familiar posture in the second test trial. However, these effects were weak, and following adjustments for multiple comparisons, they were no longer significant, suggesting that the similarly varied postures are more difficult for infants to discriminate. Also in the second test trial, the 7-month-olds’ greater gross motor development was related to significantly stronger novelty preferences for the upright headless postures than the inverted headless postures. The 9-month-olds, in the second test trial had stronger novelty preferences in the inverted whole-figure condition than the upright condition, and this was related to low and average gross and fine motor development.

## 10. General Discussion

### 10.1. Infants’ Discrimination of Body Postures

Seven-month-old infants demonstrated an ability to discriminate postures in our initial studies but only with the upright headless postures. The limb positions of the familiar and novel test images were clearly contrasting, but infants were sensitive to the differences in fairly short trials (i.e., 5 s). The infants did not discriminate postures presented in an inverted orientation or with postures presented with heads regardless of orientation. Infants spent 72% of the time focused on heads, suggesting that the presence of heads reduced the time infants needed to discriminate bodies in the whole-figure condition. However, with about-facing stimuli, where faces were not visible, infants still only discriminated postures when heads were absent. Analyses of the gaze data (see Appendix A) revealed a significant drop in attention to the heads in the about-facing study, but infants still spent more than half the time looking at the heads (59%). The focus on heads could explain the lack of the discrimination of bodies in previous studies where heads were included (e.g., [55]; [63]). Gaze data here indicated that infants focused on bodies the longest in the upright headless condition across all studies (94%), which likely contributed to their ability to recognise the difference in postures.

Once the complexity of the limb positions between the familiar and novel postures was increased in similarity, infants failed to demonstrate a sensitivity to differences in body postures even with headless upright postures with 5 s trials. With longer trial durations (i.e., 8 s), there was weak evidence for 7- and 9-month-old infants’ discrimination of upright whole figures, but these did not survive corrections for multiple comparisons. This suggests that infants can discriminate body postures, but this ability is only clearly evident with highly contrasting body stimuli.

### 10.2. Implications for Infants’ Visuospatial Body Representations

With adults, the absence of heads reduces or eliminates the body inversion effect (BIE), suggesting that typical configural processing of bodies is impeded with atypical depictions of human figures (e.g., [3]; [73]). [2] ([2]) also found that heads attracted adults’ attention during posture discrimination of upright whole postures, but this supported posture discrimination (see also [4]). In the first two studies here, the heads of the upright figures were likely a distraction for infants reducing infants’ capacity to discriminate postures, as was also found by [30] ([30]) in their study of infants’ discrimination of attractive and unattractive bodies. The lack of discrimination of upright whole-figure postures, in our first two studies, was contrary to predictions and results from [31] ([31]), who found that infants discriminated whole figures and not incomplete figures. In [31]’s ([31]) study, the largest change in the posture of the whole figures was to the arm positioned next to the head, raising questions about whether Hock et al.’s findings were also due to a focus on the upper region. The headless stimuli here were unlike the incomplete stimuli in [31]’s ([31]) study, which were also headless but were also missing a torso and some of the limbs. Our headless stimuli still contained the configuration of a body. With adults, configural processing does not necessarily require holistic presentations of stimuli, as [51] ([51]) found a BIE among adults when bodies were presented as vertical halves. However, adults require the heads for the typical configural processing of bodies, as seen with the reduced BIE with headless stimuli ([2]; [3]; [73]). Our findings suggest that the entire form might not be necessary for infants’ configural processing of bodies.

With the similarly complex body postures (Studies 2A and 2B) in the current investigation, there was weak evidence for infants’ discrimination of postures in the whole-figure condition rather than the headless condition, but only when presented with 8 s rather than 5 s trials. Longer trials possibly allowed infants to move from a focus on heads to incorporating bodies in their comparison of postures. Looking time proportions to heads and bodies did not differ significantly between the studies with 5 and 8 s trials, with more than 50% of the infants’ focus directed towards the heads in both studies. The longer trials perhaps allowed sufficient time to compare the bodies. [75] ([75]) found, with two 20 s preferential looking trials, that 9- but not 5-month-olds preferred bodies with typically over atypically proportioned body parts. In a separate study with a longer pre-familiarisation (i.e., 30 s) to a body with atypically proportioned body parts, infants as young as 3.5 months preferred the typically proportioned bodies ([77]). Therefore, trial length can be critical for infants’ capacity to demonstrate sensitivity to differences in bodies. The effect of the presence or absence of heads on infants’ discrimination of bodies might be affected by trial length.

As inversion effects are stronger with familiar categories, this implies that faces and bodies are subject to specialised visual processing (see [8]; [51]; [72]). Studies finding an FIE in infancy (e.g., [13]; [69]) suggest that the mechanisms for configural representations of upright faces exist in infancy. We found that infants discriminated body postures and in the upright conditions only. [75] ([75], [76], [77]) also found sensitivity to body structure, proportions, and emotions with upright bodies rather than inverted bodies, and this mounting evidence suggests that upright bodies are more familiar to infants and subject to specialised configural processing. However, other studies raise questions as to whether infants engage in configural processing of bodies, with [27] ([27]) finding that differential brain responses to upright and inverted bodies do not appear until 14 months of age. However, evidence of activation in body-selective areas of the brain (i.e., EBA and FBA) to images of bodies in the first year of life ([38]) and differential hemodynamic activation in the STS to upright compared to inverted dynamic bodies in 5- and 8-month-old infants ([37]) suggests that the neural mechanisms for configural processing of bodies are present.

The findings have implications for understanding infants’ social–cognitive development. Infants predominantly focus on faces but not to the detriment of interest and capacity to discriminate bodies. They largely focus on faces but also gather information from bodies, which can have implications for their social interactions with others when interpreting information from bodies, such as intentions and emotional states (e.g., [50]; [54]; [76]; [78]).

### 10.3. Relationships with Motor Development

Stronger gross motor development was linked with an inversion effect among the 7-month-olds, with stronger novelty preferences for the upright compared to the inverted headless postures. Infants focused on bodies the most in the headless upright conditions. Therefore, for younger infants, greater use of their bodies was linked with the condition where infants directed their focus on bodies the most, that is, when presented in their typical orientation (i.e., upright headless). With the 9-month-olds, lower and average levels of gross and fine motor development were linked with stronger novelty preferences for the inverted compared to the upright whole figures. When bodies were inverted, infants looked at the bodies of the whole figures more than when they were upright. Infants also focused on the feet the longest in the inverted whole and headless conditions, and the 9-month-old infants with less developmentally mature motor development were possibly more prone to an upper bias and distinguished the postures by the leg and feet positions.

Thus, we have found a body discrimination orientation effect linked with infants’ gross motor skills. Infants’ better ability to discriminate headless bodies in their familiar orientation suggests they have formed a stable representation of bodies, and this is linked to the use of their own bodies. Importantly, this was observed with stimuli where the heads were absent, suggesting that the link with gross motor development became apparent when the infants were focused on the body stimuli. When viewing whole figures, the relationship with motor development involved a reversed inversion effect with better discrimination of inverted over upright whole figures. As infants have a greater focus on heads when viewing whole figures, this is further evidence that when bodies are not the key focus, performance is diminished and less related to motor development. When whole figures are upside down, infants looked more at the bodies, which was linked with better discrimination and better motor development.

### 10.4. Implications for Sensorimotor Influences on Infants’ Visuospatial Body Representations

The findings here contrast with previous studies that did not find a relationship between motor development and infants’ capacity to discriminate bodies ([15]; [55]). The significant relationships in the current study were found when gross and fine motor scores were moderated into low, medium, and high levels and when comparing novelty preferences between the upright and inverted postures. Therefore, the effect is possibly more subtle in nature and reflects a more tentative and slowly developing relationship between infants’ visuospatial and sensorimotor representations of bodies. [55] ([55]) suggest that the ability to discriminate bodies during infancy is dependent on exposure to bodies and how close stimuli resemble bodies typically seen in everyday life. However, we found a significant relationship with the arguably less typical images of bodies. That is, 7-month-olds’ gross motor development was related to stronger novelty preferences for upright headless postures, and 9-month-olds’ fine and gross motor development was related to stronger novelty preferences for inverted whole figures. Both [15] ([15]) and [55] ([55]) presented bodies with heads, which potentially detracted attention. The legs and feet of the bodies in [55]’s ([55]) study only differed in the length and not the position, and even when inverted they were perhaps more difficult for infants to differentiate compared to the stimuli presented here. Postural stimuli are likely more familiar and relevant for infants than scrambled ([63]) or stretched depictions of infant bodies ([55]). Infants also likely have less exposure to infant bodies, like those used in [55]’s ([55]) study. Body postures reflect actions ([78]), and infants’ own embodied experience of different gross motor body positions ([68]), and previous experience attending to and imitating others’ body movements ([35]) might better relate to images of body postures.

It is likely that the link between infants’ developing visuospatial and sensorimotor body representations is evident in contexts where the ability to focus on bodies is enhanced—that is, when heads are absent or bodies are sufficiently different, as is the case with the legs and feet of inverted body postures. While the current study did not directly measure infants’ ability to integrate sensory and proprioceptive input, our findings align with the notion that infants’ sensorimotor body representations emerge as motor skills develop and influence their visual capacity to recognise and detect differences in body postures. To successfully navigate and socialise within the environment, infants rely on multisensory feedback from their body movements and visual experiences ([14]). As infants’ motor skills emerge, they become more familiar with body postures and the configuration of their body—providing infants with visual and proprioceptive sensory input related to their bodies ([68]). Multisensory input of other modalities (i.e., visual and tactile) can enhance newborn infants’ perceptual sensitivity to body-related information ([24]). When [63] ([63]) presented images of body configurations with human voices, they found that this visual and auditory input enhanced infants’ discrimination of bodies. Motor development may help infants refine their perceptual sensitivity to body postures and enrich their multisensory representations of bodies.

The type of measure of motor development is also worth considering. [55] ([55]) assessed whether or not infants could move independently (i.e., crawl or walk), and [53] ([53]) measured how dextrous infants were when playing with toys. The gross motor items of the Ages and Stages Questionnaire (ASQ-3, [66]) used here include infants’ ability to sit independently and stand or walk with support, and the fine motor items reflect infants’ ability to grasp small items and use their fingertips. Therefore, the position of the arms, legs, and feet in our stimuli were likely relevant features for infants and were related to the motor items of the ASQ-3 (see also [6]). [53] ([53]) found a relationship between infants’ fine motor development and their responses to dynamic stimuli, namely arm and hand movements. We instead found that weaker fine motor development was related to 9-month-olds’ stronger novelty preferences for the inverted compared to the upright whole figures. This is possibly due to the images of the current investigation reflecting broader limb positions rather than fine motor actions. Thus, it is possible that the relationship between infants’ visuospatial body representations and motor development is evident when the motor assessment tool is more closely linked to the stimuli.

### 10.5. Limitations and Future Directions

The measure of motor development was based on a brief screening tool, which could be too coarse for a potentially weak relationship. Future studies should use a detailed tool dedicated to measuring motor development, such as the Alberta Infant Motor Scale ([47]) and the Neuro-Sensory Motor Developmental Assessment (NSDMA; [10]). However, as mentioned, there might be key items that correspond to the type of stimuli used, so multiple approaches should be assessed.

More advanced motor skills might support a more advanced sensorimotor capacity to explore the environment and develop a visual sensitivity to bodies ([15]; [42]). An increased capacity to remain in an upright position, move independently, and hold objects at a close range are also associated with infants’ increased cognitive and perceptual sensitivity to objects ([43]; [74]). Therefore, it is unclear if relationships between infants’ capacity to discriminate bodies are related to their motor development per se or instead to their increased capacity to visually explore the world and detect differences in stimuli. Future studies should measure both cognitive and motor development to disentangle the contribution of each to infants’ sensitivities to bodies. Measuring a clinical population, such as infants with cerebral palsy, where body movements and postures are impacted might reveal the contribution of motor skills to visuospatial body representations. Measuring infants ‘at risk’ of autism (see [36]) might also be useful as autistic children and adolescents are more likely to have a delay in their integration of multisensory body information and have atypical body representations ([57]). More pronounced autistic traits can impact the ability to identify simple actions ([9]). Expanding the population of interest to include a clinical population with atypical body representations could allow future research to measure if sensorimotor representations play a critical role in an infant’s visual sensitivity to bodies.

Measuring the relationship between motor development and visual body perception based solely on visual input may provide an incomplete understanding of this relationship. It does not account for the contribution of synchronous multisensory integration, which can enhance the visual perceptual sensitivity in body-related stimuli ([24]). Therefore, it is unclear if our findings reflect the true effect motor development has on infants’ body representations. As infants achieve motor-based milestones, they are increasingly able to engage with their bodies and those of others in multisensory ways, through touch, proprioception, and coordinated movement, potentially enhancing the salience of body-specific visual information. Future research should aim to measure how synchronous multisensory integration enhances the capacity to recognise and detect differences in headless and whole-figured body postures.

The familiarisation novelty preference method has numerous challenges. One challenge is determining the appropriate trial length to sufficiently familiarise infants, which can affect whether they prefer the novel or familiar item or indeed any test item (e.g., [26]; [32]). Pilot testing contributed to the trial lengths used here. The 7-month-old infants failed to discriminate the more similarly complex postures presented in 5 s trials and demonstrated only a weak ability with 8 s trials, albeit with a familiarity preference. The 9-month-olds also demonstrated only a weak capacity to discriminate the similarly varied postures. Longer trial durations likely allowed infants to shift their attention away from the heads to the bodies, but the most suitable trial duration might differ by age. An added difference for the studies, with the similarly varied postures, is that the study with 5 s trials was conducted using an EyeLink 1000, and the study with 8 s trials was conducted with an EyeLink 1000 Plus (due to the lead authors’ relocation). Infants’ difference in performance could reflect the differing eye-tracking sampling rates (500 Hz versus 1000 Hz). However, this is less likely, as the familiarisation duration typically plays a vital role in infants’ capacity to discriminate images in studies using the familiarisation novelty preference method ([32]; [33]).

The number of test trials also raises questions (see also [26]; [32]). Aside from the about-facing study, where 7-month-old infants demonstrated significant novelty preferences in the first test trial, the 7-month-olds in the initial study and the 8 s study demonstrated preferences in the second test trial only. The 9-month-olds’ novelty preference for the upright whole postures was in the first test trial, which switched to a stronger preference for the inverted whole figure in the second test trial. Traditionally, looking times are summed over the two test trials before calculating a novelty preference (e.g., [49]), but here we saw changes in preferences across test trials. As comparative looking to the test images occurs within test trials, novelty preferences should be calculated within each trial. Future studies should consider how many test trials are necessary with a given age and stimulus type.

As there was potential evidence of the infants’ discrimination of body postures in the whole-figure condition with 8 s trials, despite a continued focus on heads, this suggests that a more global focus on bodies occurred. We analysed gaze data with distinct areas of interest, but the infants’ gaze might have fallen on the interest area borders, such as that seen by [2] ([2]) where adults focused on the heads and upper torso region. Future studies should take a more nuanced approach to measuring infants’ gaze that is less discrete and based more on pixel-level heat maps of fixations determined by the participants (see [12]; [39]). Removing entire heads for the headless stimuli might also have introduced an unnatural, attention-grabbing element to the stimuli. We attempted to address the presence of faces with the about-facing stimuli, but future studies should take a more graded approach by reducing the visibility of the head with blurring or by reducing the opacity of the head, thereby increasing the ecological validity of the stimuli.

**Conclusions.** We provide further evidence of infants’ capacity to discriminate bodies, but only upright bodies—the most visually familiar orientation. However, we provide unique evidence of the infants’ direction of gaze during body posture discrimination, revealing that infants’ demonstration of a sensitivity to differences in upright postures is constrained by the presence of heads. The degree of similarity between the test items and trial length was also influential. The findings contribute to the mounting evidence that in the first year of life infants have a configural representation of bodies presented in static images, but it is perhaps weaker than it is for faces. Infants prefer to look at faces when available, and infants’ visual experience alone may be sufficient to develop a perceptual sensitivity to faces ([15]). We also provide unique evidence that infants’ gross motor development might contribute to infants’ sensitivity to bodies, which could explain the more protracted nature of infants’ developing body representations. However, the detection of a relationship might depend on the measure of motor development being well linked to the type of body stimuli.

Bodies, like faces, are socially relevant and contribute to our recognition of identities, emotional states, gender, age, and intentions and are important for our interactions with the social and physical environment ([45]; [64]; [70]). The current study provides evidence for a connection between infants’ sensorimotor and visuospatial representations, which do not develop in isolation and both contribute to infants’ developing knowledge of people. Being able to detect information from body postures specifically can be useful in interpreting a person’s actions, intentions, and emotional states ([50]; [78]). We provide evidence that in the first year of life infants’ increased capacity to move and use their bodies could contribute to their social–cognitive development based on the information gained from observing others’ body movements. In real-life contexts, this happens dynamically, in contrast to the laboratory context, where infants view body movements in a snapshot form. As real-world body perception is based on biological motion, future studies should include dynamic stimuli, which might reveal a clearer relationship with motor development and infants’ sensorimotor representations of bodies.

## Figures and Tables

**Figure 1 behavsci-15-01021-f001:**
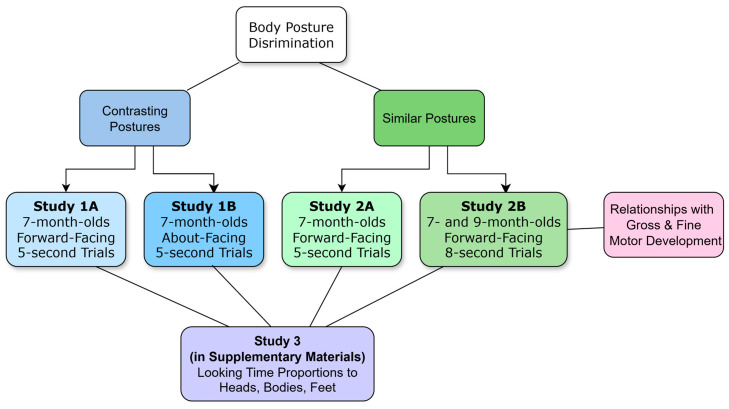
Overview of studies.

**Figure 2 behavsci-15-01021-f002:**
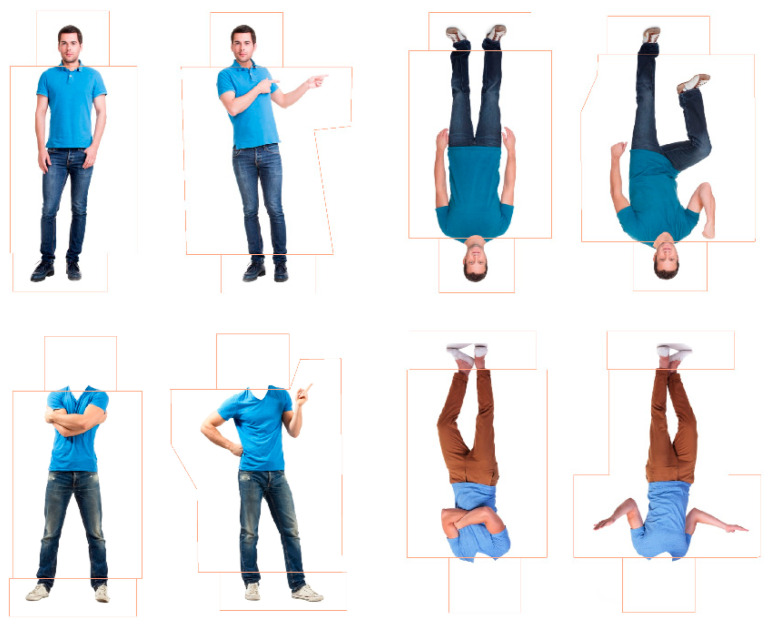
Example stimuli pairs in the whole figure upright, whole figure inverted, headless upright, and headless inverted conditions. Note. Areas of interest were not visible to the infants.

**Figure 3 behavsci-15-01021-f003:**
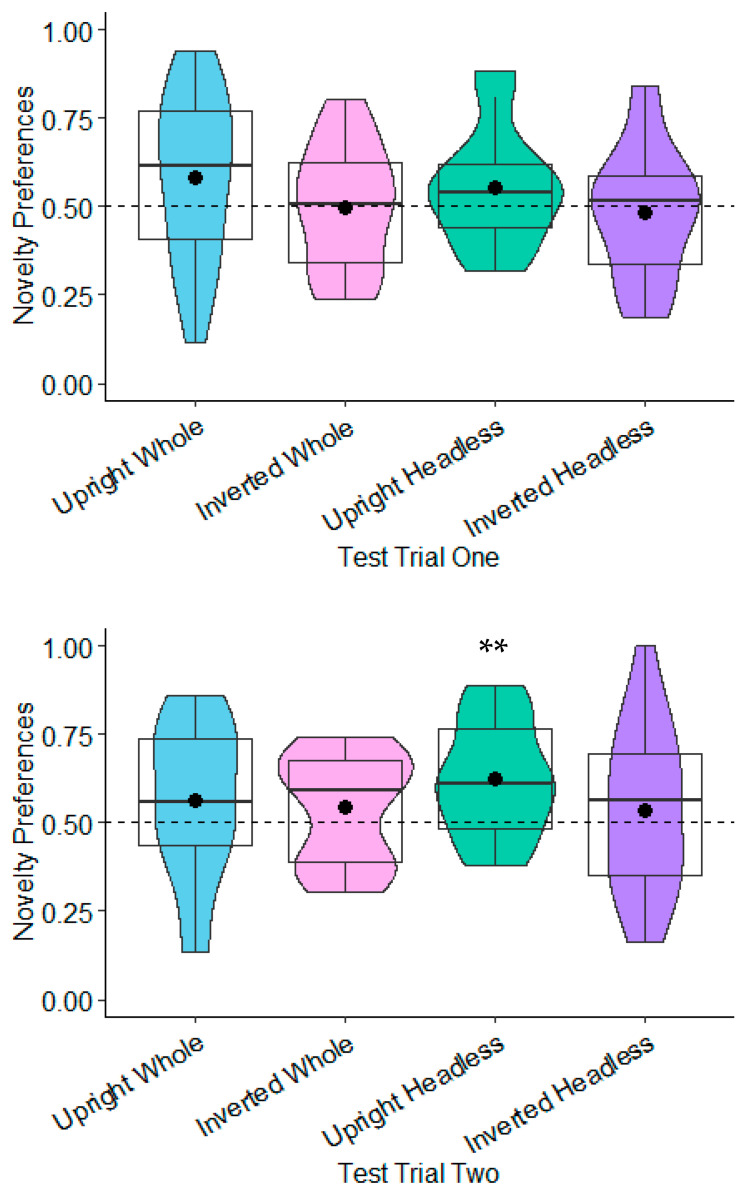
Test Trial One novelty preferences. ** *p* < 0.01.

**Figure 4 behavsci-15-01021-f004:**
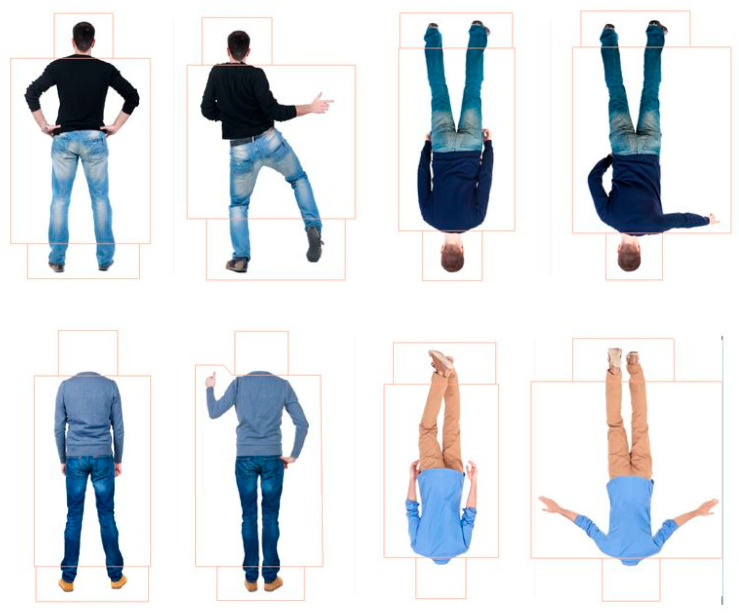
Example stimuli pairs in the whole figure upright, whole figure inverted, headless upright, and headless inverted conditions.

**Figure 5 behavsci-15-01021-f005:**
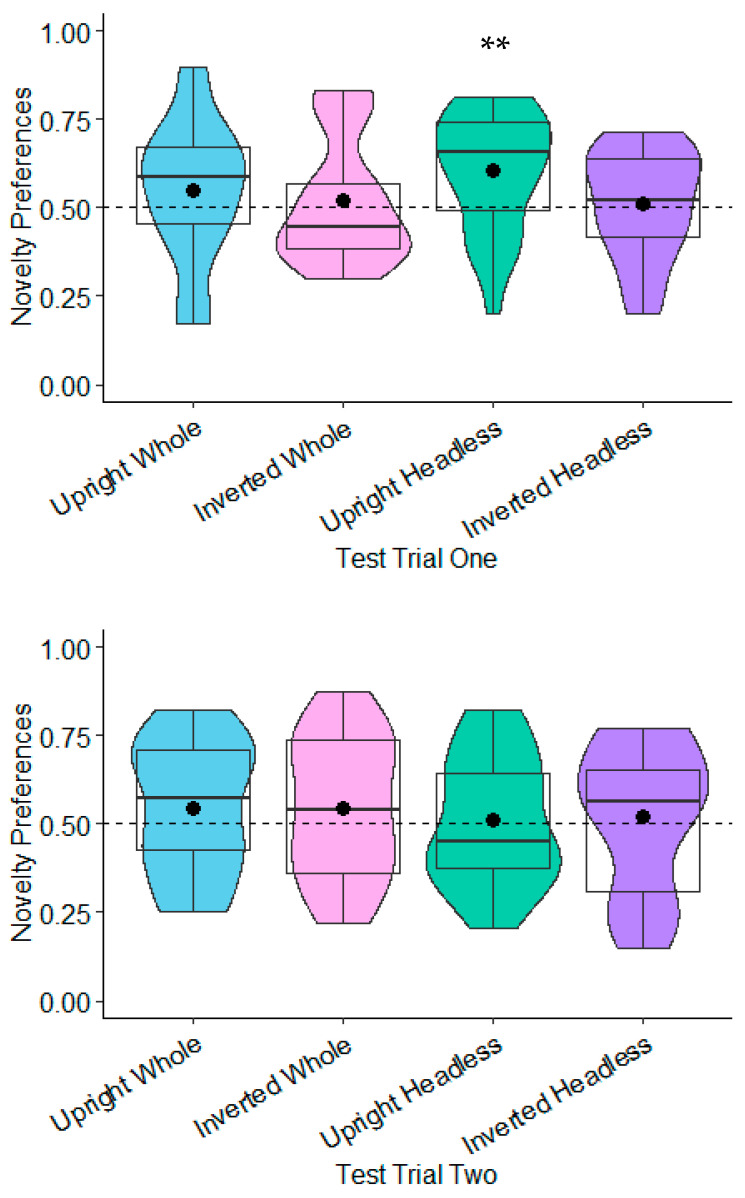
Test trial one novelty preferences with about-facing postures. ** *p* < 0.01.

**Figure 6 behavsci-15-01021-f006:**
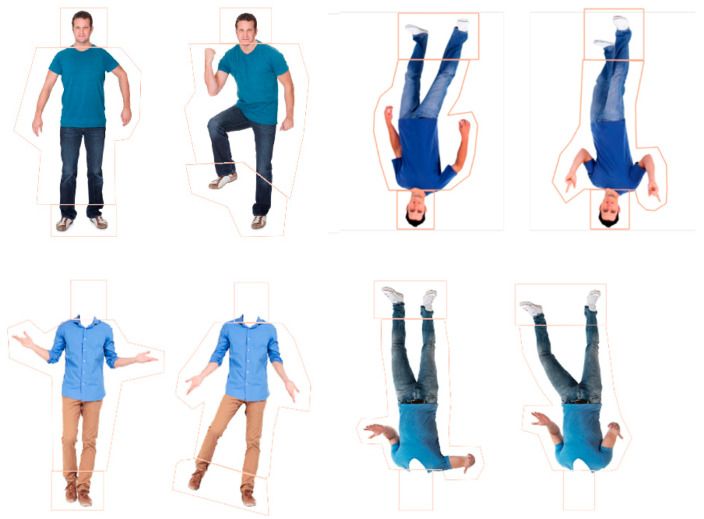
Example of areas of interest created in Data Viewer. Note. From left to right: whole figure upright, whole figure inverted, headless upright, and headless inverted.

**Figure 7 behavsci-15-01021-f007:**
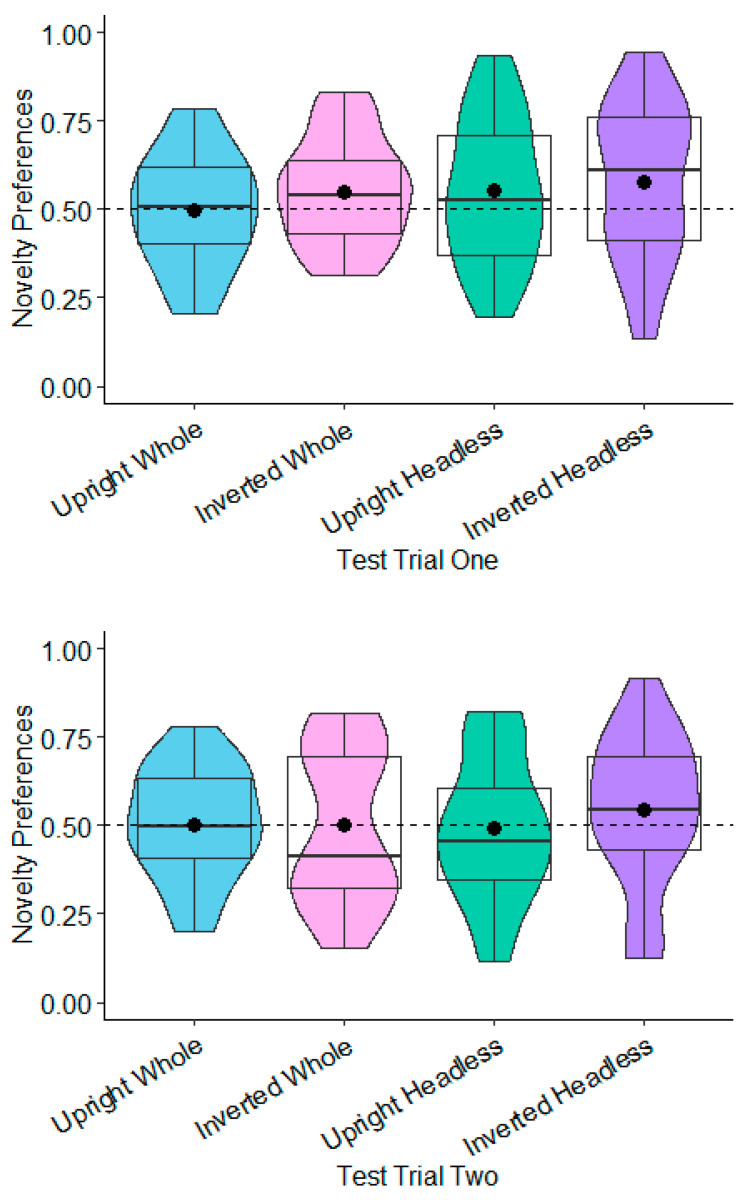
Test Trial One novelty preferences with similar-sized body postures.

**Figure 8 behavsci-15-01021-f008:**
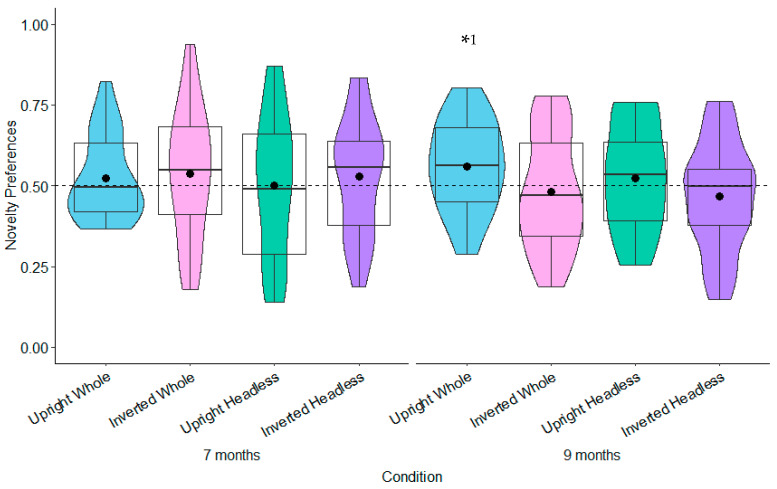
Test Trial One novelty preferences. Note. Dashed line = chance (0.50); * *p* < 0.05, ^1^ non-significant with Benjamini–Hochberg FDR adjustments.

**Figure 9 behavsci-15-01021-f009:**
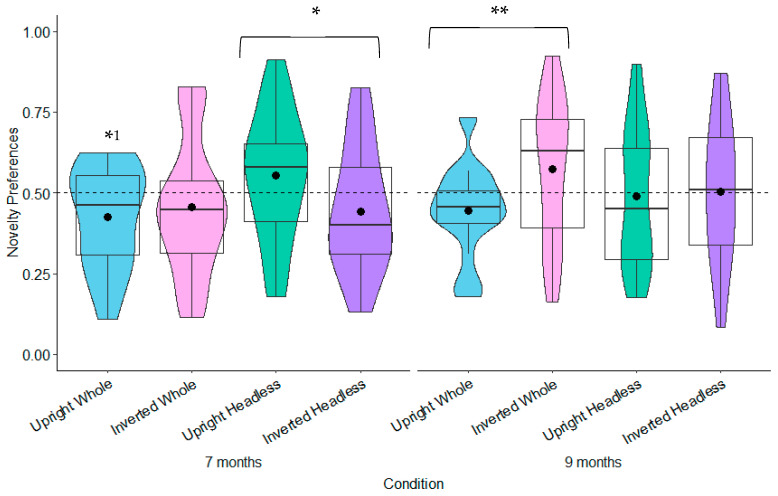
Test Trial Two novelty preferences. Note. Dashed line = chance (0.50); * *p* < 0.05; ^1^ non-significant with Benjamini–Hochberg FDR adjustments. ** *p* < 0.01.

**Figure 10 behavsci-15-01021-f010:**
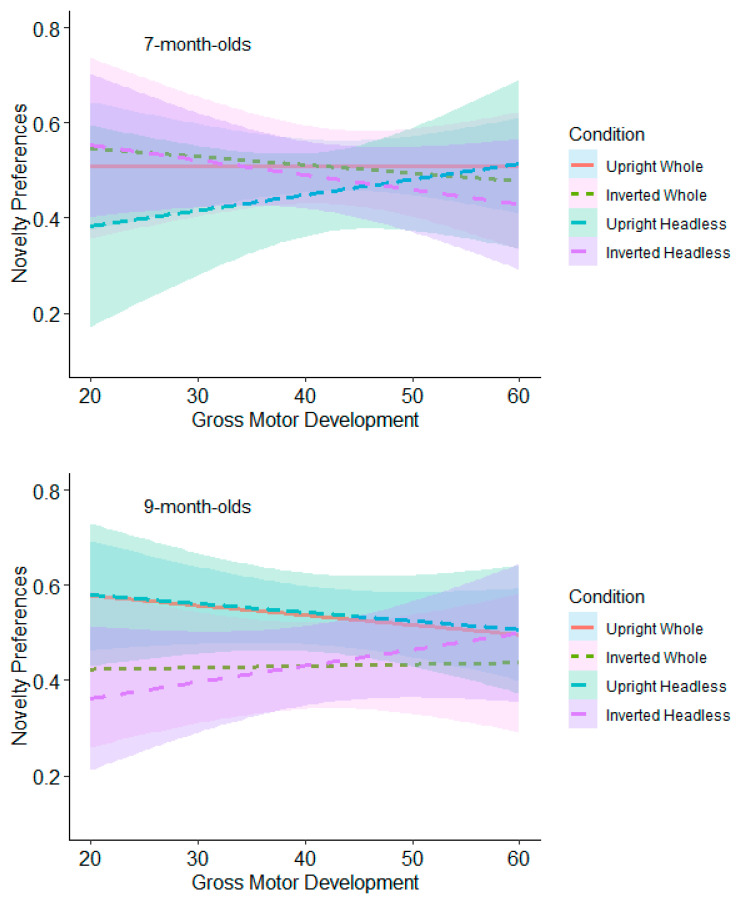
Test Trial 1: relationship between gross motor scores and novelty preferences and comparison between upright and inverted conditions.

**Figure 11 behavsci-15-01021-f011:**
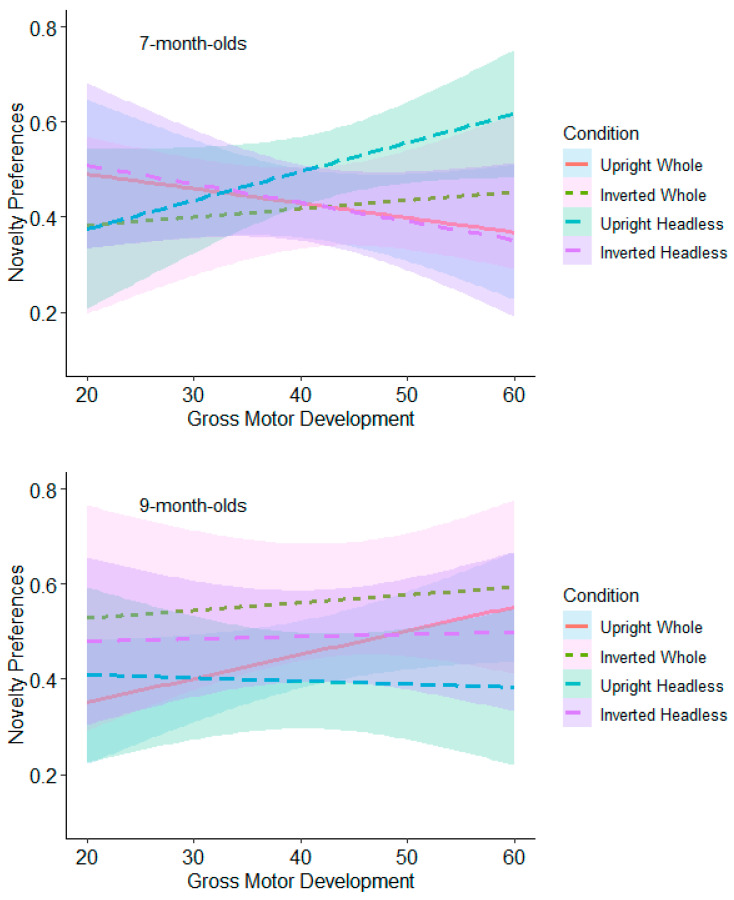
Test Trial 2: relationship between gross motor scores and novelty preferences and comparison between upright and inverted conditions.

**Figure 12 behavsci-15-01021-f012:**
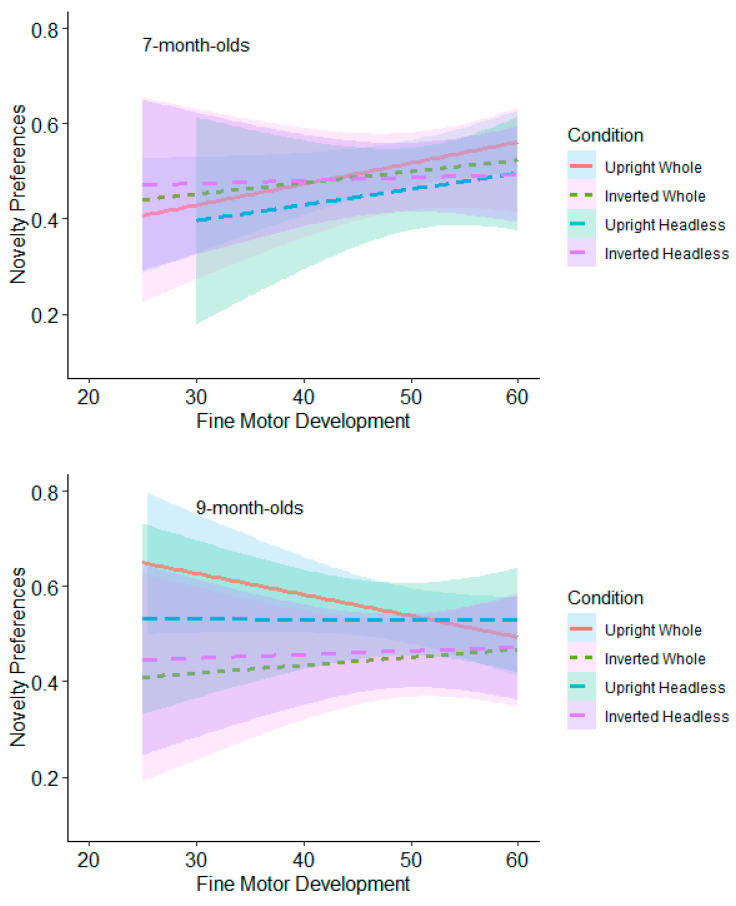
Test Trial One: relationship between fine motor scores and novelty preferences and comparison between upright and inverted conditions.

**Figure 13 behavsci-15-01021-f013:**
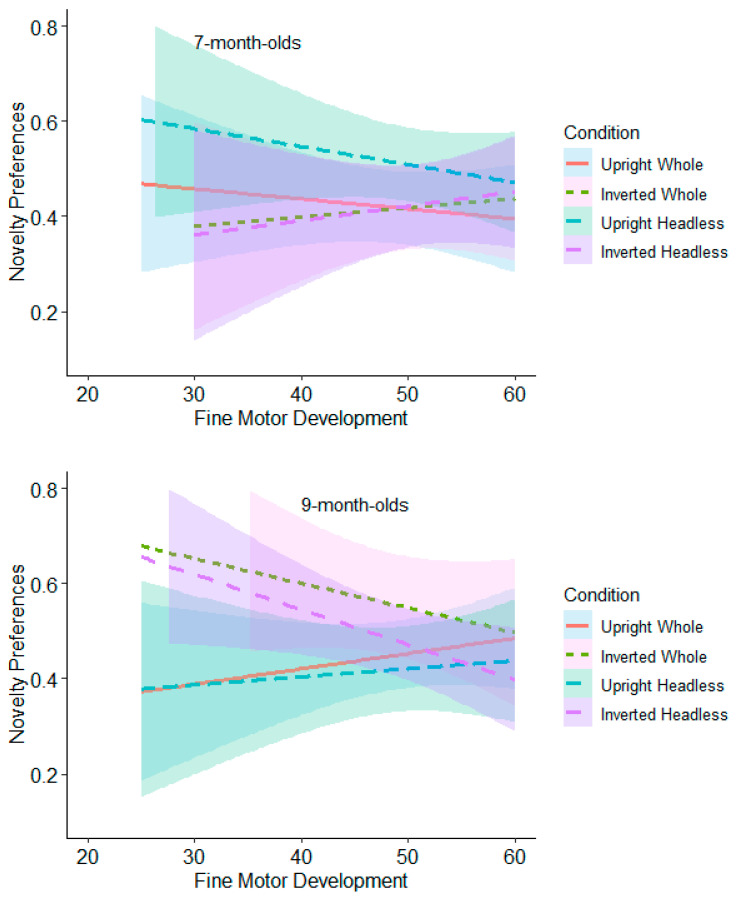
Test Trial Two: relationship between fine motor scores and novelty preferences and comparison between upright and inverted conditions.

## Data Availability

Data are available on the Open Science Framework (https://osf.io/epc5z/, URL accessed on 3 January 2025).

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
