# Peer review of "Strike a Pose: Relationships Between Infants’ Motor Development and Visuospatial Representations of Bodies"

_behavsci, 2025, doi:10.3390/bs15081021_

Round 1

Reviewer 1 Report

Comments and Suggestions for Authors

The theme that the authors address is that of the relationships between the motor development of Infants and the visuospatial representations of bodies which, in addition to being certainly interesting, is treated in depth. Their research is constantly introduced and related to the most appropriate literature and is carried out in compliance with the most accurate scientific research. The results and their analysis are carefully justified and compared consistently with the appropriate literature as well as the conclusions of the research carried out. 

The only consideration I can make is related to the presentation of the results. The data relating to the 4 studies determined by the different conditions under study determine a large quantity of numbers that, in the way in which they were presented, makes it difficult to quickly understand the phenomenon described. In the absolutely complete form in which the results were presented by the authors, the entire process carried out is certainly clear, including the careful statistical analysis. This certainly makes the study reproducible by anyone who wants to repeat it. In my opinion, however, to make the reading of the results smoother and therefore easier to understand, the authors could reduce the amount of data presented, showing above all the numbers of the most significant data. For example, 8 tables are presented with the complete data of the statistical relationships between all the observed variables and an equal number of graphs are added that represent the values ​​of the observed parameters. Could the presentation be simplified by showing only the graphs with an indication of which ones have presented significant data?

Obviously this is just my opinion which I repeat, only concerns the form of how the results are presented and this does not take anything away from the quality of the work done.

Author Response

The theme that the authors address is that of the relationships between the motor development of Infants and the visuospatial representations of bodies which, in addition to being certainly interesting, is treated in depth. Their research is constantly introduced and related to the most appropriate literature and is carried out in compliance with the most accurate scientific research. The results and their analysis are carefully justified and compared consistently with the appropriate literature as well as the conclusions of the research carried out. 

Thank you very much for your positive feedback.

Comment 1: The only consideration I can make is related to the presentation of the results. The data relating to the 4 studies determined by the different conditions under study determine a large quantity of numbers that, in the way in which they were presented, makes it difficult to quickly understand the phenomenon described. In the absolutely complete form in which the results were presented by the authors, the entire process carried out is certainly clear, including the careful statistical analysis. This certainly makes the study reproducible by anyone who wants to repeat it. In my opinion, however, to make the reading of the results smoother and therefore easier to understand, the authors could reduce the amount of data presented, showing above all the numbers of the most significant data. For example, 8 tables are presented with the complete data of the statistical relationships between all the observed variables and an equal number of graphs are added that represent the values ​​of the observed parameters. Could the presentation be simplified by showing only the graphs with an indication of which ones have presented significant data?

Obviously this is just my opinion which I repeat, only concerns the form of how the results are presented and this does not take anything away from the quality of the work done.

Response 1: Thank you for this suggestion. We agree that the linear mixed effects model and moderation analyses tables are likely too cumbersome in the main manuscript and the results are fairly well represented in the text and graphs. We have moved Tables 1-8 to the Supplementary Materials (now Tables S2, S4, S5, S6, S7, S8, S10, S12).

Reviewer 2 Report

Comments and Suggestions for Authors

see attached.

Author Response

Thank you for submitting your manuscript for review. I found the research idea presented in this paper to be novel and potentially interesting.

However, I have several concerns that need to be addressed before this manuscript can be considered for publication.

Comment 1: Significance and Focus: The significance of this study needs to be clarified and strengthened. While the research idea is novel, it is not immediately clear what the broader implications of these findings are. The paper would benefit from a more thorough discussion of how this research contributes to our understanding of infant development, motor skills, and/or visuospatial representation.

Response 1: Thank you for this suggestion. We agree that the broader picture was overshadowed somewhat by the presentation of the results.  We have endeavoured to make the significance clearer in multiple places throughout the manuscript (p. 2, lines 46-49; p. 4, lines 156-160; p. 4, lines 189-192; p. 25, lines 805-811; p. 26, lines 870-883;  pp. 28-29, lines 989-1003). The focus is on not only infants’ ability to distinguish bodies, but how their developing motor skills contributes to this.

Comment 2: Currently, the paper reads more like a thesis than a journal article. The scope is quite broad, and there is a large amount of data presented. To improve readability and focus, I strongly recommend splitting the data and corresponding aims into several papers. The methods and results sections are dense and difficult to follow due to the massive representation of data, and this affects readers to stay focused on the aims of the study.

Response 2:  Yes, we agree, and along with another reviewer’s comment, we have moved all the tables to the Supplementary Materials as the text and graphs sufficiently represent the key findings. We have also moved Study 3 (gaze data comparisons to heads, bodies, feet in all the studies) to the Supplementary Materials. Study 3 helps with interpretating the body discrimination findings, but are not essential in the main manuscript. We believe the manuscript is now more readable and easier to digest. We would have liked to have created more papers, but the first two studies (Studies 1A and 1B) seemed insufficient on their own and the third and fourth studies (2A and 2B) addressed the limitations of the first two studies (e.g., the contrasting familiar and novel body shapes). The motor development data is essential for this special issue as it links motor and cognitive development (visuospatial representations of human bodies).

Results and Conclusion:

Comment 3: The conclusion is also too short to adequately summarize the substantial amount of information presented in the results section. It needs to be expanded to provide a more comprehensive synthesis of the findings.

Response 3: See also Comments 1, 4, & 5, where we further spell out the conclusions of the findings, but we have also further clarified the findings of an orientation effect and the relationship with motor development on p. 25, lines 826-837. p. 26, lines 870-883.

Comment 4: Furthermore, the paper needs to clearly articulate what useful information can be gathered from the findings. This should go beyond simply stating the results and should instead explain the implications of the findings for future research, theory, or practice.

Response 4: We further articulated the implications of the findings along with further research ideas on p. 27, lines 924-944 and on p. 29, lines 1001-1004

Recommendations:

To improve this manuscript, I recommend the following:

Comment 5: Clarify the significance of the study by explicitly stating its contribution to the field.

Response 5: We now highlight in the conclusion that we are the first known study to use eye tracking revealing that infants’ direction of gaze can affect the findings (p. 28, lines 975-977). We are also the first known study to find a relationship between infants' gross motor development and infants' sensitivity to body information in static images (p. 28, lines 983-985). In the previous draft, we stated how examining body postures might be more familiar for infants (relative to the stimuli used in previous studies; see p. 4. Lines 183-188). Previously we also stated the use of eye tracking in this research (p. 5, lines 208-219). Previously, on p. 26, line 840-841 we highlighted that we find a relationship between motor development and discrimination of bodies which previous studies did not find.

Comment 6: Refocus the paper by splitting it into several more concise manuscripts with specific aims.

Streamline the methods and results sections to improve readability and focus.

Response 6: With the large tables and eye tracking/gaze data analyses moved to the Supplementary Materials we believe the main focus of the paper is clearer, particularly with regards to the special issue topic – ‘The Role of Early Sensorimotor Experiences in Cognitive Development’. The focus is on infants’ sensitivity to differences in bodies and how their motor development relates to this ability. The eye tracking data helps with interpretation of infants’ looking responses and putting the gaze data in the Supplementary Materials should help with readability. The contribution of the gaze data is that there is a large focus on heads when they are present and greater looking at bodies when heads are absent.

  • Expand the conclusion to provide a more comprehensive summary of the findings. See Comments 3 and 5.
  • Clearly discuss the implications of the findings and their potential applications. See Comment 4.

Addressing these concerns will help to improve the clarity, focus, and impact of your research. I encourage you to rewrite and resubmit your manuscript after carefully considering these comments.

We thank you for these suggestions. They have helped us to spell out the broader picture of the research and sell the key findings.

Reviewer 3 Report

Comments and Suggestions for Authors The present study contributes to understanding of infant discrimination of bodies/postures and moves forward an area that has mixed findings.  The researchers also incorporate a measure of gross and fine motor skills. The study uses eye-tracking for infant responses.  The design seems appropriate for infants aged 7 to 9 months.  The ASQ is used to measure fine and gross motor development.  The ASQ is frequently used in research and is an appropriate measure. Overall, the study makes an important contribution, has a sound design and the manuscript is well presented. I have a few queries and areas where corrects are required. It would be helpful to have an image showing an infant participating in the study to see the positioning, for example. Clothing colour of the inverted headless example in the Figures (2,4,6) is quite different to the other conditions and there are variations in clothing across conditions.  Was this variation counterbalanced across presentations? The classifications for the ASQ are correctly described lines 506-508.  These are: on schedule, further monitoring needed, further assessment required.  These classifications are later labelled low, medium and high, which is incorrect.  The label 'high' suggest better than average rather than 'on schedule', for example.  I also wondered if there was any discussion with parents about the classifications that indicated further monitoring or assessment was required. The moderation analyses included the three categories from the ASQ.  The related figures (e.g. Figure 10) seem to be using the ASQ raw score, not the categories and therefore doesn't help with understanding the analysis.  Ethnicity of most participants is listed as Australian, but that’s a nationality, not an ethnicity.  It looks as though the list for participants has a mix of ethnicity and nationality. There are multiple mentions of the research as ‘ethically approved by the human research ethics committee..’  The sentence does not need ‘ethically’.

Author Response

The present study contributes to understanding of infant discrimination of bodies/postures and moves forward an area that has mixed findings.  The researchers also incorporate a measure of gross and fine motor skills. The study uses eye-tracking for infant responses.  The design seems appropriate for infants aged 7 to 9 months.  The ASQ is used to measure fine and gross motor development.  The ASQ is frequently used in research and is an appropriate measure. Overall, the study makes an important contribution, has a sound design and the manuscript is well presented.

Thank you for the positive feedback. We are pleased it can make a contribution to the field.

Comment 1:  I have a few queries and areas where corrects are required. It would be helpful to have an image showing an infant participating in the study to see the positioning, for example.

Response 1: We do not have ethical approval to share images of infants taking part, but EyeLink (SR-Research) have a clear image of the set-up. Our infants were in a darkened room, but I have added a link referring readers to the image. https://www.sr-research.com/eyelink-1000-plus/

Comment 2: Clothing colour of the inverted headless example in the Figures (2,4,6) is quite different to the other conditions and there are variations in clothing across conditions.  Was this variation counterbalanced across presentations?

Response 2: Yes, the people being presented (out of 8 possible exemplars) was counterbalanced across the conditions. We have clarified this in the Procedure section (p. 7 lines 316-317). In the figures, there are 4 of each as examples, but oddly the headless inverted examples were inadvertently the ones with lighter clothing. To balance this, I swapped the headless upright and inverted examples in Figure 6.

Comment 3: The classifications for the ASQ are correctly described lines 506-508.  These are: on schedule, further monitoring needed, further assessment required.  These classifications are later labelled low, medium and high, which is incorrect.  The label 'high' suggest better than average rather than 'on schedule', for example. The moderation analyses included the three categories from the ASQ.  The related figures (e.g. Figure 10) seem to be using the ASQ raw score, not the categories and therefore doesn't help with understanding the analysis. 

Response 3: The values for the clinical ASQ classifications differ to the values for the levels in the moderation analyses as these were based on the raw scores in the current sample with low referring to scores 1 standard deviation (SD) below the mean of the sample, high referring to those with scores 1 SD above the mean, and medium referring to scores within 1 SD of the mean. We have clarified this in the Method section (on p. 15, lines 548-550) and in the Results section where we first report a moderation analysis (on p. 19, line 641). For the scores themselves, they are in the tables, but I have added them to the text for the significant results.

Comment 4: I also wondered if there was any discussion with parents about the classifications that indicated further monitoring or assessment was required.

Response 4: Yes, as part of the ethical approval process, we did recommend to caregivers to contact their general practitioner or child and family health nurse when scores were in the lowest range where further assessment with a professional is recommended.

Comment 5: Ethnicity of most participants is listed as Australian, but that’s a nationality, not an ethnicity.  It looks as though the list for participants has a mix of ethnicity and nationality.

Response 5:  The sociodemographic questionnaire we used had allowed the participants to self-assign the ethnicity they identify with using an open-ended question. The Australian Bureau of Statistics do recognise Australian as an ethnicity https://www.abs.gov.au/statistics/classifications/australian-standard-classification-cultural-and-ethnic-groups-ascceg/latest-release

I have reworded the text to reflect the fact that the participants self-assigned their ethnicities (p. 6, lines 252-253; p. 9 line 372; p. 12 line 437; pp. 14-15 lines 504-505).

For future studies, we will use the recent recommendations from a publication by the Many Babies consortium who have outlined a series of questions to aid in categorising participants’ ethnicities https://www.tandfonline.com/doi/full/10.1080/15248372.2024.2431106

Comment 6: There are multiple mentions of the research as ‘ethically approved by the human research ethics committee.’  The sentence does not need ‘ethically’.

Response 6: Thank you for detecting this. This has been corrected (p. 6, line 237; p14, line 485).

Reviewer 4 Report

Comments and Suggestions for Authors

This study’s key strength lies in its careful, multi-angle examination of how 7- and 9-month-old infants discriminate human body postures. Conducting several cross-sectional eye-tracking experiments with such a challenging population is impressive, and the authors are to be commended for clearly reporting ethics approval, open-data availability, and a priori sample-size calculations. Because I am not an expert in infant eye-tracking theory, my comments concentrate on methodology and analysis. I hope my comments will be helpful to the authors.

First, it is difficult to see exactly which findings are genuinely robust. Although the Limitations and Conclusion sections touch on this issue, questions remain about the degree of experimental control. Removing the entire head produces an unnatural stimulus; a more graded manipulation, such as blurring or lowering head opacity, would likely be more ecologically valid. Moreover, the study relies solely on static images, omitting the biological-motion cues that are crucial to real-world body perception. If static stimuli are retained, a richer analysis would help—for instance, the supplementary file contains ROI bar and line graphs, but pixel-level fixation heat maps would have been far more informative.

Second, several design changes are confounded. The 5-s / 7-month / ANU (EyeLink 1000) condition and the 8-s / 7 + 9-month / Newcastle (EyeLink 1000 Plus) condition differ simultaneously in exposure duration, participant age, hardware, and laboratory environment, making it hard to identify which factor drives the outcome differences; the apparent 5-s versus 8-s effect could even reflect the higher sampling precision of the newer eye-tracker. Reanalyzing the data with lab or equipment included as random effects might clarify this issue.

Third, extreme novelty scores (< 0.05 / > 0.95) were deleted, and values beyond +-2 z were replaced with the nearest score. This trimming and imputation scheme risks distorting effect sizes and feels somewhat ad hoc.

Finally, the paper reports a very large number of statistical tests across Experiments 1A–2B, yet I found no description of any adjustment for multiple comparisons, even though the design is repeated-measures. One-sample or paired t-tests appear to have been treated as if the conditions were independent, and after the LMM each three-way interaction is unpacked with more t-tests. If corrections such as Bonferroni, Holm, or FDR were applied, please kindly state this.

Author Response

This study’s key strength lies in its careful, multi-angle examination of how 7- and 9-month-old infants discriminate human body postures. Conducting several cross-sectional eye-tracking experiments with such a challenging population is impressive, and the authors are to be commended for clearly reporting ethics approval, open-data availability, and a priori sample-size calculations. Because I am not an expert in infant eye-tracking theory, my comments concentrate on methodology and analysis. I hope my comments will be helpful to the authors.

Thank you very much for the positive feedback and for the thoughtful comments to help improve the paper.

Comment 1: First, it is difficult to see exactly which findings are genuinely robust. Although the Limitations and Conclusion sections touch on this issue, questions remain about the degree of experimental control. Removing the entire head produces an unnatural stimulus; a more graded manipulation, such as blurring or lowering head opacity, would likely be more ecologically valid. Moreover, the study relies solely on static images, omitting the biological-motion cues that are crucial to real-world body perception. If static stimuli are retained, a richer analysis would help—for instance, the supplementary file contains ROI bar and line graphs, but pixel-level fixation heat maps would have been far more informative.

Response 1: Thank you very much for these ideas. We have expanded on the limitations on p. 27 (lines 962-966) and the last paragraph p. 28.

Comment 2: Second, several design changes are confounded. The 5-s / 7-month / ANU (EyeLink 1000) condition and the 8-s / 7 + 9-month / Newcastle (EyeLink 1000 Plus) condition differ simultaneously in exposure duration, participant age, hardware, and laboratory environment, making it hard to identify which factor drives the outcome differences; the apparent 5-s versus 8-s effect could even reflect the higher sampling precision of the newer eye-tracker. Reanalyzing the data with lab or equipment included as random effects might clarify this issue.

Response 2: This is a good question about the possible reasons for the different findings. With the statistics software (jamovi), I was unable to successfully run the original LMM analysis with ‘eye tracker type’ included (Study 3 that has data from the two locations). This was possibly due to the unequal numbers across locations. I then reduced the analysis to just compare the 7-month-olds who saw the similarly-sized postures tracked with the EyeLink 1000 (with 5-second trials) and the 7-month-olds who were tracked with the EyeLink 1000 Plus (with 8-second trials). The ‘eye tracker’ type variable was non-significant, t(52.77) = -0.66, p = .509. Note, this analysis did not run properly either as it would not calculate results for the interactions. We have however, added the eye tracker factor to the Discussion, but given that effects of trial duration are typical with familiarisation novelty preference methods (Houston-Price & Nakai, 2004; Hunter & Ames, 1988), this is a likelier explanation. The results were also largely the same with the similarly-sized stimuli for the 7-month-olds tested with the EyeLink 1000 and the 7-month-olds tested with the EyeLink 1000 Plus (particularly following adjustment for multiple comparisons). This is discussed on p. 27, lines 936-943.

Comment 3: Third, extreme novelty scores (< 0.05 / > 0.95) were deleted, and values beyond +-2 z were replaced with the nearest score. This trimming and imputation scheme risks distorting effect sizes and feels somewhat ad hoc.

Response 3: The first step of removing novelty preferences of <0.05 and >0.95 is standard with the familiarisation novelty preference method (e.g., Quinn & Eimas, 1998) and necessary because it suggests the infant did not give sufficient attention to both images in a test trial in order to form a preference. The second step of then assessing for values more than 2 standard deviations from the mean is important to ensure the data is normally distributed and results are not based on extreme scores. I have clarified our intentions with these steps on p. 8, lines 328-332.

Comment 4: Finally, the paper reports a very large number of statistical tests across Experiments 1A–2B, yet I found no description of any adjustment for multiple comparisons, even though the design is repeated-measures. One-sample or paired t-tests appear to have been treated as if the conditions were independent, and after the LMM each three-way interaction is unpacked with more t-tests. If corrections such as Bonferroni, Holm, or FDR were applied, please kindly state this.

Response 4: For the one-sample t-tests compared to chance, we have adjusted the critical p-values using the FDR/Benjamini-Hochberg method (e.g., Yekutieli & Benjamini, 1999). The effects for the first two studies (Studies 1A & 1B) remained significant. However, for Study 2B (the previously significant effects are no longer significant pointing the robustness of the findings with the similarly varied postures. More importantly, the effect sizes were small and given the disagreement in the literature on adjusting critical p-values (e.g., Althouse, 2016; Barnett et al., 2022; Feise, 2002), we have focused on the effect sizes. This has been updated in the Abstract, Results, and the Discussion on the following pages.

Results:

  • 8, lines 341-344
  • 10, lines 397-400
  • 15, lines 574-577
  • 16, lines 582-585
  • 22, lines 712-714

Discussion:

  • 23, lines 740-744
  • 23, line 769 and 771
  • 26-27, lines 932-934
  • 27, line 955

Relevant updates were also noted in the Supplementary Materials.

For the follow-up simple effects for the LMMs, most of these were to understand the significant interactions, but we have adjusted them using the FDR/Benjamini-Hochberg method (e.g., Yekutieli & Benjamini, 1999). The jamovi software automatically includes some comparisons that are unnecessary and not relevant to our research questions (e.g., HLU vs WFI). We were only interested in the upright vs. inverted comparisons within each body type condition (headless or whole figure) so we have removed these unnecessary contrasts in the follow-up tests.

We have adjusted the text accordingly on the following pages.

  • 18, lines 629-631 and lines 636-638
  • 19, lines 655-657 and lines 662-664
  • 20, lines 678-680
  • 21, paragraph 1

Reviewer 5 Report

Comments and Suggestions for Authors

The topics discussed in the article are interesting and useful as research objects. The background of research, proposed solutions, obtained results, statistical analysis, discussion, conclusions, and limitations were adequately and accurately described. Summing up, the paper is professionally written and therefore I recommend the submitted manuscript for publication after making a few minor corrections following the comments below.

1) I recommend using the term Areas of Interest (AOIs) instead of Interest Areas (IAs), as the former is used in scholarly literature on eye tracking.

2) I suggest writing the phrase "eye tracking" separately without a hyphen while "eye-tracking camera" or "eye-tracking data" should be hyphenated.

3) For improved structure and readability of the article, I suggest numbering the chapters and adding an introductory section titled Introduction at the beginning, to organize the content more clearly.

Author Response

The topics discussed in the article are interesting and useful as research objects. The background of research, proposed solutions, obtained results, statistical analysis, discussion, conclusions, and limitations were adequately and accurately described. Summing up, the paper is professionally written and therefore I recommend the submitted manuscript for publication after making a few minor corrections following the comments below.

We appreciate the positive feedback, and many thanks for the extra suggestions.

Comment 1: 1) I recommend using the term Areas of Interest (AOIs) instead of Interest Areas (IAs), as the former is used in scholarly literature on eye tracking.

Response 1: These have been changed throughout.

Comment 2: 2) I suggest writing the phrase "eye tracking" separately without a hyphen while "eye-tracking camera" or "eye-tracking data" should be hyphenated.

Response 2: Yes, we agree and have changed this throughout.

Comment 3: 3) For improved structure and readability of the article, I suggest numbering the chapters and adding an introductory section titled Introduction at the beginning, to organize the content more clearly.

Response 3: Thank you for this suggestion. These have added to the clarification of the break-up of the studies.

Round 2

Reviewer 2 Report

Comments and Suggestions for Authors

Dear Editor

Thank you for sending me this revised manuscript for a second review. I appreciate you considering my expertise again.

However, after a quick preliminary look at the revised submission, it appears that the core issues I raised in my initial review regarding clarity, conciseness, and overall polish largely remain unaddressed. Given that my initial recommendation was for rejection due to these fundamental concerns, I regret that I must respectfully decline to provide a second detailed review at this stage. I believe the manuscript would benefit from a more substantial revision and professional editing before it is ready for re-evaluation by reviewers.

I apologize for any inconvenience this may cause and wish you the best in finding an alternative reviewer.

Author Response

Read

Reviewer 4 Report

Comments and Suggestions for Authors

The authors responded to my comments with sincerity and incorporated them to the fullest extent, for which I am grateful. All of my questions have been resolved. I would also like to extend my personal encouragement and appreciation for the research team’s scholarly dedication in completing this long-term study.